# ARID1A governs the silencing of sex-linked transcription during male meiosis in the mouse

Debashish U Menon, Prabuddha Chakraborty, Noel Murcia, Terry Magnuson*

Department of Genetics, and Lineberger Comprehensive Cancer Center, The University of North Carolina at Chapel Hill, Chapel Hill, United States

## eLife assessment

This study presents a **valuable** dataset regarding chromatin remodeling by the BAF complex in the context of meiotic sex chromosome inactivation. **Solid** data generally support the conclusions, although the partial deletion of the BAF complex in the germline could be considered limiting. This work will be of interest to researchers working on chromatin and reproductive biology.

*For correspondence:
tmagnuson@unc.edu

Competing interest: The authors declare that no competing interests exist.

## Abstract

We present evidence implicating the BAF (BRG1/BRM Associated Factor) chromatin remodeler in meiotic sex chromosome inactivation (MSCI). By immunofluorescence (IF), the putative BAF DNA binding subunit, ARID1A (AT-rich Interaction Domain 1 a), appeared enriched on the male sex chromosomes during diplonema of meiosis I. Germ cells showing a Cre-induced loss of ARID1A arrested in pachynema and failed to repress sex-linked genes, indicating a defective MSCI. Mutant sex chromosomes displayed an abnormal presence of elongating RNA polymerase II coupled with an overall increase in chromatin accessibility detectable by ATAC-seq. We identified a role for ARID1A in promoting the preferential enrichment of the histone variant, H3.3, on the sex chromosomes, a known hallmark of MSCI. Without ARID1A, the sex chromosomes appeared depleted of H3.3 at levels resembling autosomes. Higher resolution analyses by CUT&RUN revealed shifts in sex-linked H3.3 associations from discrete intergenic sites and broader gene-body domains to promoters in response to the loss of ARID1A. Several sex-linked sites displayed ectopic H3.3 occupancy that did not co-localize with DMC1 (DNA meiotic recombinase 1). This observation suggests a requirement for ARID1A in DMC1 localization to the asynapsed sex chromatids. We conclude that ARID1A-directed H3.3 localization influences meiotic sex chromosome gene regulation and DNA repair.

## Introduction

Meiosis is central to the formation of haploid gametes from diploid progenitors. During meiosis, homologous chromosomes pair, exchange genetic material, align at the metaphase plate, and then segregate during the first meiotic cell division. Unlike autosomal homologs, the X and Y chromosomes share limited homology. Unique mechanisms have evolved to ensure these chromosomes become sequestered and processed in a parallel but distinct process from autosomes.

XY chromosomes pair along the pseudo-autosomal region (PAR), leaving extensive regions of the X and Y unpaired. These regions become enriched for DNA damage response (DDR) factors, including ATR (ataxia telangiectasia and Rad3 related), TOPBP1 (DNA topoisomerase II-binding protein 1), and MDC1 (mediator of DNA damage checkpoint 1), which amplifies γH2A.X across XY chromatin (*Alavattam et al., 2021*; *Ellnati et al., 2017*; *Fernandez-Capetillo et al., 2003*; *Ichijima et al., 2011*;

*Royo et al., 2013*; *Turner et al., 2005*). XY chromatin acquires further distinguishing features when the variant histone H3.3 replaces the canonical histones H3.1/3.2, and H3K9me3 (histone lysine9 trimethylation) becomes elevated on XY chromatin relative to autosomes (*Hirota et al., 2018*; *van der Heijden et al., 2007*; *Yuen et al., 2014*). As the XY chromosomes become epigenetically distinct from autosomes, they also become physically separated into a unique nuclear sub-compartment known as the XY body (*Handel, 2004*). Transcriptional silencing of unpaired chromatin occurs during this phase of meiosis I, known as meiotic silencing of unpaired chromatin (MSUC; *Turner, 2007*). MSUC of autosomal chromatin, such as in response to an abnormal karyotype, causes pachynema arrest and cell death (*Baarends et al., 2005*; *Schimenti, 2005*; *Turner et al., 2005*). However, silencing unpaired XY chromatin results in a different outcome known as meiotic sex chromosome inactivation (MSCI). Sex chromosomes are required to undergo MSCI and become transcriptionally inactivated. Failure to achieve MSCI triggers arrest and cell death (*Ichijima et al., 2011*).

Chromatin factors that influence MSCI include the H3K9 methyltransferase, SETDB1 (SET domain, bifurcated 1), PRC1 associated SCML2 (Sex comb on midleg-like protein 2), and testis-specific reader of histone acetylation, BRDT (bromodomain testis-specific protein; *Hasegawa et al., 2015*; *Hirota et al., 2018*; *Manterola et al., 2018*). Previous studies demonstrated a critical requirement for SWI/SNF-directed transcriptional regulation during meiosis (*Kim et al., 2012*; *Menon et al., 2019*; *Wang et al., 2012*). The germ-cell-specific depletion of the SWI/SNF catalytic subunit, BRG1 (Brahma-related gene-1), resulted in an early pachytene arrest (*Kim et al., 2012*). BRG1 is associated with the XY body (*Wang et al., 2012*), but its requirement for MSCI is unknown. This lack of evidence for MSCI is also true for the PBAF (POLYBROMO1- BRG1/BRM associated factor) remodeler within the SWI/SNF family, which we have shown is required for meiotic cell division (*Menon et al., 2021*). Whether these data suggest a role for PBAF in MSCI remains an open question, given that BRG1 associates with both SWI/SNF remodelers, PBAF and BAF.

Here, we find that ARID1A (AT-rich interaction domain 1 a), a BAF-specific putative DNA binding subunit, is required for the transcriptional silencing of the sex chromosomes during pachynema, thereby ensuring meiotic progression through prophase-I. Mechanistically, ARID1A is necessary to establish MSCI hallmarks, such as the eviction of elongating RNA polymerase-II (phosphorylated on Serine2 of carboxy-terminal domain; pSer2-RNAPII), limited sex-linked chromatin accessibility, and hyper-accumulation of histone H3.3 on the sex body. Additionally, ARID1A is required for the targeted localization of DMC1 to the unpaired sex chromatids, implicating BAF-A-governed sex-linked chromatin dynamics in DNA repair.

## Results

### Meiotic progression requires ARID1A

Meiotic progression in male mice depends on SWI/SNF-regulated gene expression (*Menon et al., 2019*; *Menon et al., 2021*). These studies raise the prospect of distinct SWI/SNF subcomplexes governing stage-specific meiotic transcription (*Menon et al., 2021*). To understand the meiotic functions of the biochemically distinct SWI/SNF BAF subcomplex, we examined the spermatogenic expression profiles of the BAF subunits *Arid1a* and *Arid1b* using single-cell RNA-seq data generated from testes (*Ernst et al., 2019*). *Arid1a* mRNA expression occurred at various spermatogenic stages with notable expression at pachynema of meiotic prophase I (*Figure 1—figure supplement 1A–C*).

*Arid1b* mRNA went undetectable until late in spermiogenesis (*Figure 1—figure supplement 1D–E*). Consistent with its mRNA expression profile, pachytene spermatocytes featured abundant ARID1A protein, which was uniformly spread among autosomes and sex chromosomes (*Figure 1*). Late in diplonema, ARID1A appeared preferentially enriched on the sex chromosomes as compared to autosomes (*Figure 1*). These data implicate ARID1A in MSCI, a process essential for meiotic progression in males.

To determine whether meiotic spermatocytes require ARID1A, we generated a germ cell-specific knock-out of *Arid1a* using the *Stra8-Cre* transgene (Stra8-C Tg/0) expressed in spermatogonia (*Sadate-Ngatchou et al., 2008*). *Stra8-Cre* was chosen among the available germ-cell-specific Cre lines to circumvent *Arid1a* haploinsufficiency during embryonic development. *Stra8-Cre* remains inactive during embryonic development when transmitted through females (*Sadate-Ngatchou et al., 2008*). The number of undifferentiated spermatogonia that stained positive for Promyelocytic leukemia zinc

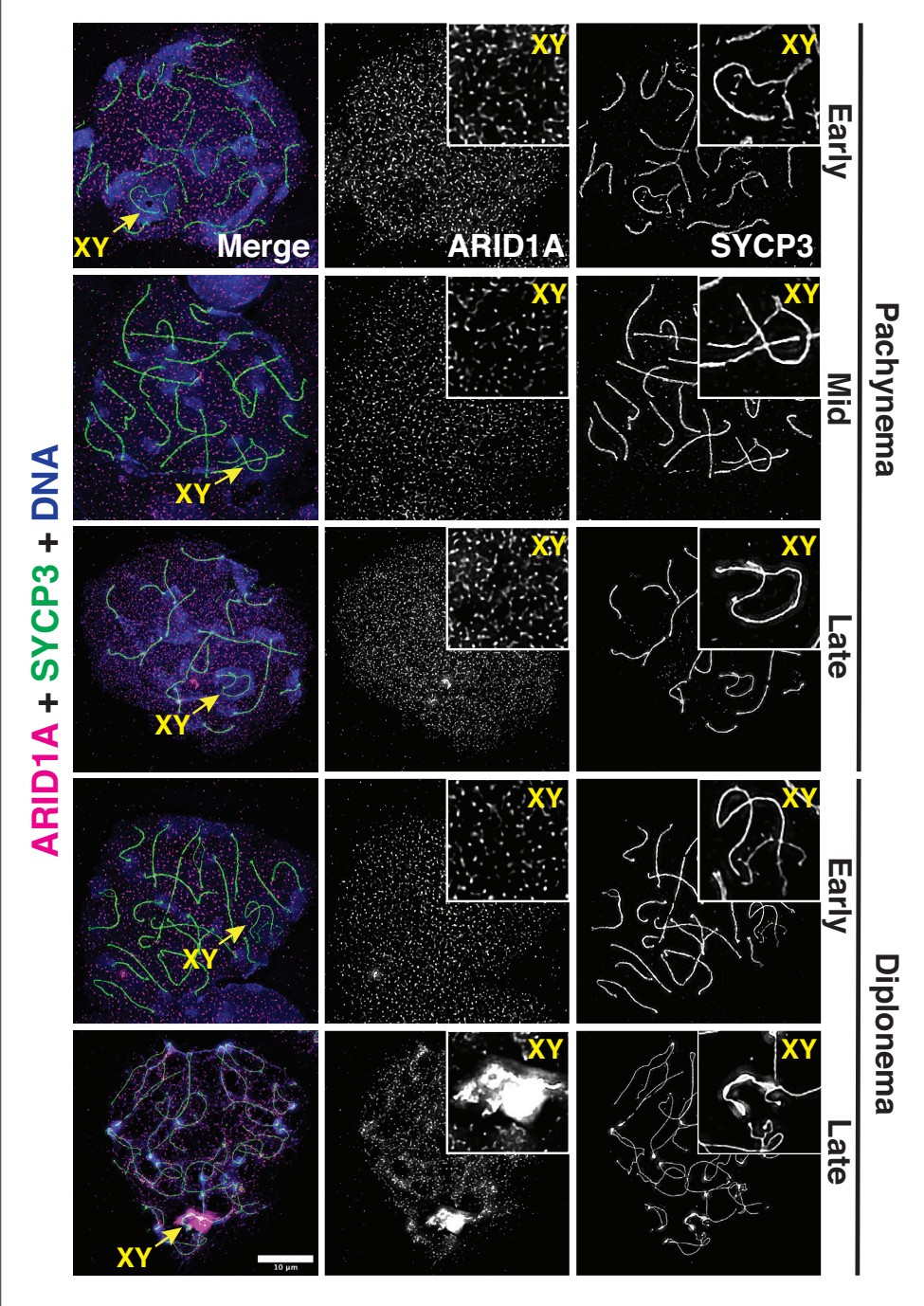

**Figure 1.** ARID1A associates with the sex body late in meiotic prophase-I. Representative wild-type pachytene and diplotene spermatocyte spreads immunolabelled for ARID1A (magenta) and SYCP3 (green) and then counterstained for DNA (blue). Panel insets denote a magnified view of the nuclear area surrounding the sex chromosomes. Scale bar:10 μm, magnification: ×100.

The online version of this article includes the following figure supplement(s) for figure 1:

**Figure supplement 1.** ARID1A transcripts detected in pachytene spermatocytes.

finger (PLZF[+], also known as Zinc finger and BTB domain containing 16: ZBTB16) remained unchanged in *Arid1a* cKO testes (*Figure 2—source data 1*). The proportions of undifferentiated spermatogonia (PLZF+) with detectable (ARID1A+) and non-detectable (ARID1A-) levels of ARID1A protein determined by immunostaining on testes cryosections obtained from 1 month old *Arid1a*[fl/fl] (control) and

*Arid1a* cKO (CKO) males was 74% ARID1A⁻ (CKO, n=179) and 26% ARID1A⁺ (CKO, n=65) as compared to 95% ARID1A⁺ (n=88) and 5% ARID1A⁻ (n=5) in controls. Western blot analysis also showed a significant reduction in ARID1A protein in mutant pachytene spermatocytes (*Figure 2—figure supplement 1*). Therefore, the *Arid1a* conditional knockouts generated with a *Stra8-Cre* did not appear to impact earlier stages of spermatogenesis. Potential defects early in spermatogenesis might occur with an earlier-acting germline Cre transgene. In this case, an inducible Cre transgene would be needed, given the haploinsufficiency associated with *Arid1a*.

Contrary to our expectations, meiosis appeared unperturbed, as evidenced by seminiferous tubules featuring round and elongated spermatids in *Arid1a* cKO testes (*Figure 2—figure supplement 2A*). Furthermore, *Arid1a* cKO epididymis featured mature spermatozoa, suggesting that *Arid1a* cKO testes are capable of normal spermatogenesis (*Figure 2—figure supplement 2B*). Although these data indicate that ARID1A is dispensable for spermatogenesis, we wanted to confirm that these results were not a technical artifact arising from inefficient *Stra8-Cre* mediated excision of the *Arid1a*ᶠˡ/ᶠˡ alleles.

To address this question, we examined *Arid1a* transcripts isolated from *Arid1a* WT and *Arid1a* cKO Sta-Put purified populations of pachytene spermatocytes and round spermatids using RT-PCR (*Figure 2—figure supplement 3A*). For cDNA synthesis, we used primers that amplify a 612 bp region spanning the *Arid1a* floxed exons 5 and 6. Assuming 100% CRE efficiency, we would expect to observe a 281 bp cDNA product associated with mRNA isolated from *Arid1a* cKO spermatogenic cells. Instead, we observed cDNAs representative of the floxed (fl) allele and, to a greater extent, the excised (Δ) allele from mRNA isolated from *Arid1a* cKO pachytene spermatocytes. These results indicated inefficient CRE activity (*Figure 2—figure supplement 3A*). More importantly, the data suggest that a fraction of pachytene spermatocytes in *Arid1a* cKO testes fail to undergo CRE-mediated excision, resulting in escapers. An examination of ARID1A levels in testes cryosections and meiotic spreads by immunofluorescence (IF) revealed a heterogenous population of pachytene spermatocytes consisting of mutants lacking and escapers expressing ARID1A (*Figure 2—figure supplement 3B and C*). On average, 70% of pachytene spermatocytes were observed lacking detectable levels of ARID1A signal by IF.

Furthermore, examining the meiotic profile associated with *Arid1a* cKO revealed an accumulation of mutant spermatocytes (ARID1A⁻), some at leptonema and zygonema and a majority at mid-pachynema relative to *Arid1a* WT spermatocyte spreads (*Table 1*). Some diplotene spermatocytes displaying a moderate reduction in ARID1A levels occurred in *Arid1a* cKO relative to *Arid1a* WT meiotic spreads and represent escapers due to inefficient *Stra8-Cre* activity (*Table 1*- bottom two lines with asterisks, also *Figure 2—figure supplement 3B and C*). However, diplotene and metaphase-I spermatocytes lacking ARID1A protein by IF were undetectable in the *Arid1a* cKO testes (*Figure 2—figure supplement 3B*). These results indicate that those spermatocytes lacking ARID1A arrest at the mid-pachytene stage. The spermatocytes that escape Cre activity progress to subsequent meiotic

**Table 1.** Meiotic profile of P23 the wild type floxed and the conditional mutant allele.
The loss of ARID1A results in a pachytene arrest. Table outlining the distribution of P23 *Arid1a* WT and *Arid1a* cKO meiotic prophase-I profiles. SYCP3 staining determined meiotic staging. Co-staining of ARID1A identified mutant spermatocytes from escapers. The total number of spermatocytes scored included 166 for *Arida*ᶠˡ/ᶠˡ and 124 for *Arid1a* cKO. *denotes *Arid1a* cKO diplotene spermatocytes with partially reduced but not complete loss of ARID1A signal relative to controls.

| Meiotic profile (P23) | *Arid1a*ᶠˡ/ᶠˡ (n=166) | *Arid1a* cKO (n=124) |
|---|---|---|
| Pre-Lep/Leptonema | 3.6% | 13.7% |
| Zygonema | - | 15.3% |
| Early Pachynema | 13.8% | 12.9% |
| Mid Pachynema | 22.8% | 40.3% |
| Late Pachynema | 7.2% | 7.2% |
| Early Diplonema | 28.5% | 4.8%* |
| Late Diplonema | 14.5% | 5.6%* |

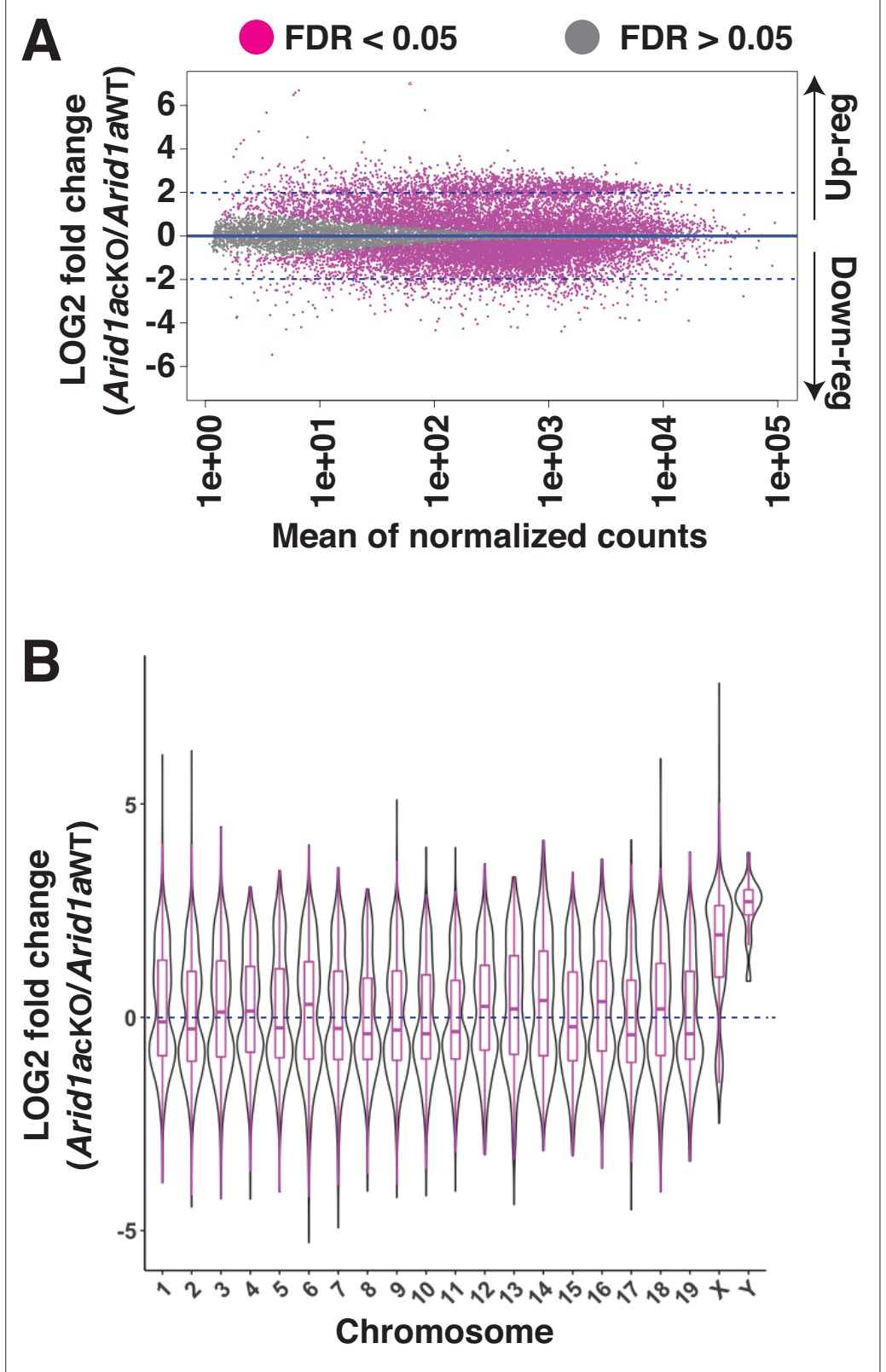

**Figure 2.** Requirement of ARID1A for the repression of sex-linked genes. (**A**) MA plot describing the LOG2-fold-change (LFC, y-axis) in mean expression (x-axis) of genes displaying significant (FDR ≤0.05, magenta dots) changes in *Arid1a* cKO relative to *Arid1a* WT pachytene spermatocytes. Dashed blue lines denote 2 LFC. Gray dots (FDR ≥0.05) depict non-significant changes in gene expression. (**B**) Violin plot describing the LOG2-fold-change (LFC,

*Figure 2 continued on next page*

*Figure 2 continued*

y-axis) in the median chromosome-wide gene expression (x-axis) in *Arid1a* cKO relative to *Arid1a* WT pachytene spermatocytes. The dashed blue line denotes no change in gene expression.

The online version of this article includes the following source data and figure supplement(s) for figure 2:

**Source data 1.** Wild-type and mutant PLZF positive spermatogonia.

**Source data 2.** ARID1A DESeq2 differentially expressed genes.

**Figure supplement 1.** Western blot for ARID1A in wild-type and mutant pachytene spermatocytes (left) and with quantitation (right).

**Figure supplement 1—source data 1.** Raw western blot images for ARID1A and nucleolin.

**Figure supplement 1—source data 2.** Complete and marked western blot images for ARID1A and nucleolin.

**Figure supplement 2.** Spermatogenesis appears unperturbed in *Arid1a* cKO males.

**Figure supplement 2—source data 1.** ARID1A tubule staging: Staining and morphology determined seminiferous tubule staging (*Ahmed and Rooij, 2009*; *Meistrich and Hess, 2013*).

**Figure supplement 3.** *Arid1*a cKO testes display inefficient *Stra8-Cre* activity.

**Figure supplement 3—source data 1.** Raw images of DNA electrophoresis gels for the analysis of the RT-PCR of *Arid1a* transcripts from STA-PUT purified populations of *Arid1a* WT and *Arid1a* cKO pachytene spermatocytes and round spermatids (*Figure 2—figure supplement 3a*).

**Figure supplement 3—source data 2.** Complete and marked images of DNA electrophoresis gels presented in *Figure 2—figure supplement 3a*.

**Figure supplement 4.** Arid1a mutant testes exhibit aberrant meiotic prophase I, reducing the abundance of round spermatids during the first wave of spermatogenesis.

**Figure supplement 5.** Transcription start sites of differentially expressed autosomal and sex-linked genes display ARID1A occupancy.

**Figure supplement 6.** ARID1A does not influence sex body formation.

---

stages, giving rise to genotypically normal gametes. All round spermatids isolated from *Arid1a* cKO testes appeared only to express the normal transcript associated with the floxed allele (*Figure 2—figure supplement 3A*). Furthermore, our evaluation of the first round of spermatid development based on DNA content (1 C, 2 C, and 4 C), revealed a significantly reduced abundance of round spermatids (1 C) in mutant testes compared to wild-type testes. This finding, obtained through flow cytometry, supports the observed meiotic block at the pachytene stage (*Figure 2—figure supplement 4A–B*). Therefore, the perceived lack of a phenotype is an artifact of inefficient *Stra8-Cre* activity. But more importantly, the data indicate that ARID1A is essential for progression beyond pachynema.

## ARID1A regulates the transcriptional silencing of the sex chromosomes at pachynema

The meiotic requirement of ARID1A and, more importantly, its association with the sex chromosomes at diplonema prompted us to examine its role in MSCI. We hypothesized that ARID1A might play a role in the transcriptional silencing of the sex-linked genes during meiosis. To address this question, we performed RNA-seq on Sta-Put purified populations of pachytene spermatocytes to profile changes in transcript abundance upon ARID1A deficiency. Differential analysis of gene expression using DeSeq2 (*Love et al., 2014*) revealed an equal proportion of significantly (FDR <0.05) misexpressed genes displaying either elevated (up-regulated, n=5824) or reduced (down-regulated, n=5821) transcript abundance in *Arid1a* cKO relative to *Arid1a* WT pachytene spermatocytes (*Figure 2A* and *Figure 2—source data 2*). Notably, we detected significant misexpression of 53% of all the sex-linked coding genes (593/1105) in response to an ARID1A deficiency. Amongst these, 86.4% displayed elevated expression (n=512), whereas only 13.6% displayed reduced (n=81) transcript abundance. This skew was even greater when only considering sex-linked genes misexpressed by a magnitude of 2 LOG-fold or higher. Here, 97% (297/306) of the misexpressed sex-linked genes displayed increased transcript abundance in response to an ARID1A deficiency. Therefore, ARID1A predominantly affects the repression of sex-linked genes during pachynema.

An examination of changes in the average transcript abundance on a chromosome-wide basis showed that expression from the sex chromosomes was significantly higher than that from autosomes

in response to the loss of ARID1A at pachynema (*Figure 2B* and *Figure 2—source data 2*). Therefore, ARID1A regulates the transcriptional repression of the sex chromosomes, implicating it in MSCI. ARID1A's influence on sex-linked gene regulation may result from its association with the sex chromosomes, particularly during diplonema, where ARID1A hyper-accumulates on the XY. However, during pachynema, when MSCI initiates, ARID1A associates with the XY at autosomal levels as detected by IF (*Figure 1*). Therefore, it is possible that, at the onset of pachynema, local ARID1A association rather than chromosome-wide coating dictates its role in sex-linked gene repression.

To address ARID1A's association with the XY chromosomes, we determined its genomic localization at a higher resolution by CUT&RUN (cleavage under targets & release using nuclease) in pachytene spermatocytes. We detected ARID1A occupancy at promoters of differentially regulated target genes, irrespective of their chromosomal location, in *Arid1a* WT relative to *Arid1a* cKO (negative control) pachytene spermatocytes (*Figure 2—figure supplement 5A*). This result is consistent with the genome-wide distribution of ARID1A observed in pachytene spermatocyte spreads by IF (*Figure 1*). Although not detectable by IF, pachytene spermatocytes displayed a preferential association of ARID1A with promoters of normally repressed sex-linked genes, when detected by CUT&RUN (*Figure 2—figure supplement 5B*), emphasizing its role in MSCI.

## ARID1A does not influence DNA damage response signaling on the sex body

Unlike autosomal homologs that complete pairing during pachynema, the non-homologous regions of the sex chromosomes feature unrepaired DNA double-strand breaks (DSBs). These DNA DSBs recruit γH2Ax, a product of DNA damage response (DDR) signaling pathways essential for establishing and maintaining MSCI (*Turner, 2007*). Therefore, to determine whether ARID1A regulated MSCI by influencing DDR signaling, we first monitored the association of γH2Ax with the sex chromosomes in response to the loss of ARID1A. By IF, γH2Ax accumulation on the sex body appeared unperturbed in the mutant (ARID1A⁻) relative to escapers (ARID1A⁺) and *Arid1a* WT pachytene spermatocytes (*Figure 2—figure supplement 6A*). Consistent with this, the recruitment of ATR (ataxia telangiectasia and Rad3 related), a kinase known to phosphorylate γH2Ax and initiate MSCI (*Royo et al., 2013*; *Turner et al., 2004*), appeared normal in the absence of ARID1A. These data indicate that sex-linked DDR signaling occurred independently of ARID1A (*Figure 2—figure supplement 6B*). As expected, MDC1, a known γH2Ax reader that is essential for sex body formation (*Ichijima et al., 2011*), remained associated with the asynapsed sex chromosomes upon ARID1A deletion during pachynema (*Figure 2—figure supplement 6C*). These data indicate that the failure to silence sex-linked genes in response to a loss in ARID1A occurred in the presence of major DDR signaling factors known to govern sex body identity. It is possible that DDR signaling recruits BAF-A to the sex chromosomes.

## ARID1A limits RNAPII localization to the sex chromosomes during pachynema

Apart from γH2Ax, the association of the sex chromosomes with RNA polymerase II (RNAPII) distinguishes the sex body from the autosomes during pachynema. Normally, pachytene spermatocytes display reduced levels of RNAPII on the sex body relative to autosomes (*Khalil et al., 2004*). This sub-nuclear localization of RNAPII coincides with increased transcriptional output, which peaks at diplonema (*Ernst et al., 2019*), underscoring the importance of targeted mechanisms regulating sex chromosome repression. Therefore, we were curious to test whether ARID1A influenced the nuclear localization of RNAPII during pachynema. We performed IF to monitor the localization of the actively transcribing (elongating) form of RNAPII marked by Serine2 phosphorylation (pSer2) on its carboxy-terminal domain (*Noe Gonzalez et al., 2021*). We monitored elongating RNAPII (pSer2) because its pausing regulates meiotic transcription (*Alexander et al., 2023*). By co-staining for RNAPII (pSer2) and SYCP3, we could stage pachynema and identify the sex chromosomes. While RNAPII (pSer2) localization to the autosomes appeared similar, we noticed increased levels of RNAPII (pSer2) association with the sex chromosomes in *Arid1a* cKO relative to *Arid1a* WT mid and late pachytene spermatocytes (*Figure 3A*). By quantifying the RNAPII (pSer2) signal generated by IF (*Figure 3—source data 1*), we detected a 1.5-fold increase (p=0.0003) in the average RNAPII (pSer2) fluorescence associated with the sex chromosomes in *Arid1a* cKO relative to *Arid1a* WT pachytene spermatocytes (*Figure 3B*).

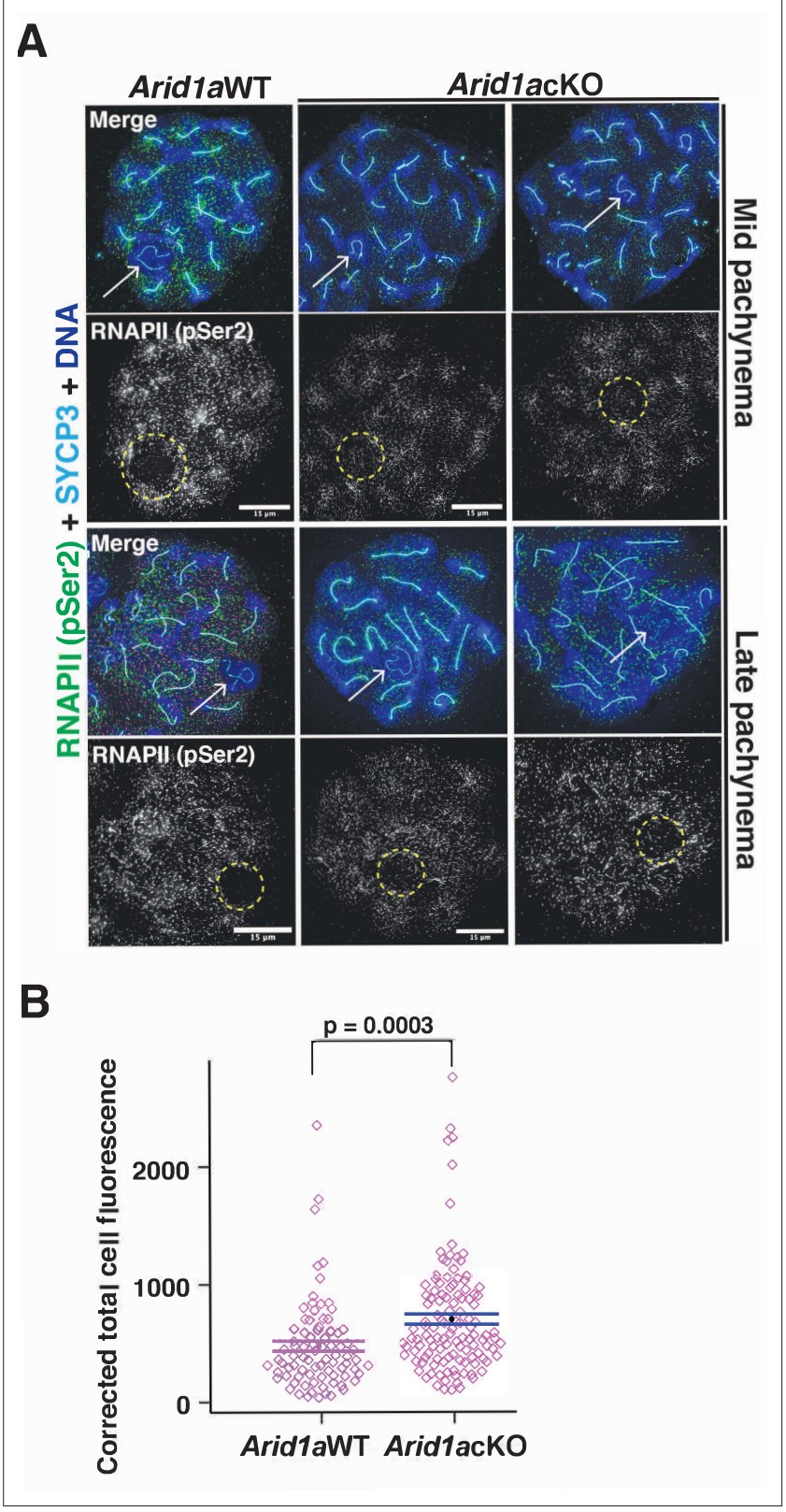

**Figure 3.** ARID1A limits RNA polymerase II (RNAPII) localization to the sex body. (**A**) *Arid1a* WT and *Arid1a* cKO pachytene spermatocytes immunolabelled for pSer2-RNAPII (green), SYCP3 (cyan), and counterstained with DAPI (blue). Scale bar:15 µm, magnification: ×100. The sex chromosomes (white arrow) and sex body (yellow dashed circle) are labeled. Brightness of the whole image panel (**A**) was increased by compressing the dynamic range of

*Figure 3 continued on next page*

*Figure 3 continued*

the image panel using Adobe photoshop (0-1-255 to 0-1-150). (**B**) Dot plot describing the corrected total pSer2-RNAPII fluorescence (y-axis) measured from *Arid1a* WT (n=82) and *Arid1a* cKO (n=119) pachytene spermatocytes (3 replicates per genotype). Empty diamonds (magenta) represent independent data points. Significance determined by a two-tailed unpaired Student's t-test p values. Data expressed as mean (black dot) ± SEM.

The online version of this article includes the following source data for figure 3:

**Source data 1.** Quantitation of pSer2RNAPII.

Due to technical difficulties, we could not simultaneously stain for ARID1A alongside SYCP3 and RNAPII (pSer2), making distinguishing escapers from mutants in the *Arid1a* cKO meiotic spreads impossible. This result suggests that the 1.5-fold increase in RNAPII association with the sex body in *Arid1a* cKO relative to *Arid1a* WT pachytene spermatocytes is likely underestimated. Therefore, our results indicate that ARID1A facilitates sex-linked gene repression by limiting the association of elongating RNAPII with the sex chromosomes during pachynema.

## ARID1A regulates promoter accessibility during pachynema

Next, we hypothesized that ARID1A-governed chromatin remodeling might underlie its role in limiting RNAPII accessibility on pachytene sex chromosomes. Therefore, we profiled changes in chromatin

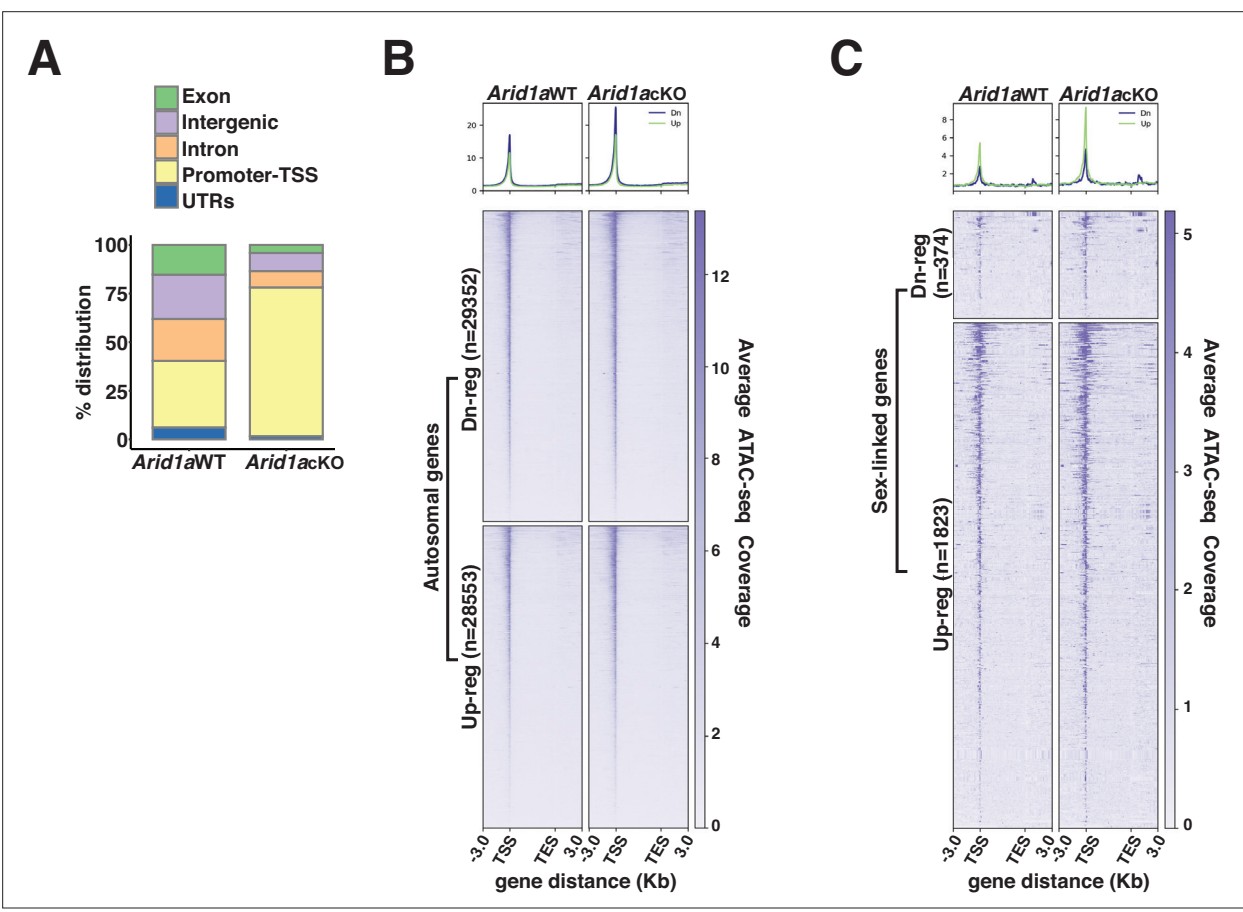

**Figure 4.** ARID1A limits promoter accessibility. (**A**) Genomic associations of MACS2 derived ATAC-seq peak calls from *Arid1a* WT and *Arid1a* cKO pachytene spermatocytes. Percent (%) distribution of genomic annotations is indicated. (**B–C**) Heatmap (bottom) and metaplot (top) displaying the average ATAC-seq signal associated with differentially expressed (**B**) autosomal and (**C**) sex-linked genes in *Arid1a* WT and *Arid1a* cKO pachytene spermatocytes. (**B–C**) The number of RefSeq annotations (n) associated with differentially regulated genes (rows) is indicated. Up-reg: misexpressed, and Dn-reg: misrepressed genes in response to the loss of ARID1A. Average ATAC-seq coverage was plotted across RefSeq genes ± 3 Kb.

The online version of this article includes the following source data for figure 4:

**Source data 1.** ATACseq peaks changes in chromatin accessibility in response to the loss of ARID1A using ATAC-seq.

accessibility in response to the loss of ARID1A using ATAC-seq. Normally, ATAC-seq peaks (accessible chromatin) are mapped comparably across promoters (34%), intergenic (22.8%), and intronic (21.5%) regions. In contrast, the loss of ARID1A resulted in a dramatic shift towards predominantly promoter-associated ATAC-seq peaks with only a minority mapping to intergenic and intronic regions (*Figure 4A* and *Figure 4—source data 1*).

Consistent with an increase in the proportion of promoter-associated peaks, we detected an increase in accessibility across transcription start sites (TSS) associated with both autosomal and sex-linked differentially expressed genes (DEGs) in *Arid1a* cKO relative to *Arid1a* WT pachytene spermatocytes (*Figure 4B and C*). Furthermore, in the case of autosomal targets, chromatin accessibility at their TSSs was enhanced irrespective of their transcriptional status, appearing indistinguishable between down-regulated and up-regulated genes upon the loss of ARID1A (*Figure 4B*). In contrast, on the sex chromosomes, the loss of ARID1A seemed to have a prominent effect on TSS's associated with up-regulated genes, which displayed greater chromatin accessibility relative to their down-regulated counterparts (*Figure 4C*). Therefore, the increased promoter accessibility of normally repressed sex-linked genes may underlie their persistent transcription by RNAPII upon the loss of ARID1A. Overall, these data highlight a role for BAF complexes in limiting promoter accessibility, especially on the sex chromosomes during pachynema.

## ARID1A regulates the chromatin composition of the sex body

Chromatin remodelers regulate DNA accessibility by altering nucleosome positioning or composition (*Clapier et al., 2017*). The latter outcome is interesting in the context of MSCI, given that the sex body typically displays a striking enrichment of the variant histone H3.3 while concomitantly appearing depleted of the canonical histones H3.1/3.2 (*van der Heijden et al., 2007*; *Yuen et al., 2014*). Given that human ARID1A regulates H3.3 genomic associations (*Reske et al., 2022*), we tested whether a similar mechanism governed H3.3 localization to the sex chromosomes. To address this, we monitored H3.3 localization in pachytene spermatocytes by IF, simultaneously staining for ARID1A and HORMAD1. The former aided in mutant identification, while the latter labeled the pachytene sex chromosomes. Consistent with previous reports, H3.3 preferentially coats the sex chromosomes, distinguishing it from autosomal chromatin during normal pachynema (*Figure 5A*). Interestingly, this sex-body association of H3.3 was detected less frequently early in pachynema (17%), becoming more pervasive during the mid (70 %) and late (95%) stages of pachynema. In contrast, in the absence of ARID1A, H3.3 staining on the sex chromosomes was indistinguishable from their autosomal counterparts throughout pachynema (*Figure 5A*). Notably, the majority of mutant mid (89%) and late (77%) pachytene spermatocytes lacked the typical sex body enrichment of H3.3 (*Figure 5A*), indicating that ARID1A dictates the preferential accumulation of H3.3 on the sex body.

Next, we were curious to know the consequence of reduced H3.3 association on H3.1/3.2 occupancy on the sex body in response to the loss of ARID1A. We performed IF to monitor the association of H3.1/3.2 with the sex chromosomes marked by HORMAD1 in response to the loss of ARID1A. Like H3.3, canonical H3.1/3.2 appeared uniformly distributed genome-wide in most early pachytene spermatocytes (92%). At the onset of pachynema, sex chromosomes mostly displayed comparable H3.3 and H3.1/3.2 levels (*Figure 5A and B*). However, this balance changed during the subsequent sub-stages of pachynema. Although robust H3.3 levels remained, H3.1/3.2 appeared excluded from the sex chromosomes in most mid (56%) and all late pachytene spermatocytes (*Figure 5A and B*). In contrast, the loss of ARID1A increased the proportion of mid (66%) and late (40%) pachytene spermatocytes retaining H3.1/3.2 on sex-linked chromatin (*Figure 5B*). The abnormal retention of canonical H3.1/3.2 coincides with a lack of H3.3 enrichment on sex-linked chromatin in response to the loss of ARID1A.

To determine whether these abnormal kinetics result from a genome-wide deficiency in H3.3, we compared its levels by western blotting to find no difference in H3.3 abundance in *Arid1a* cKO relative to *Arid1a* WT spermatocyte enriched populations (*Figure 5—figure supplement 1A*). We also considered an alternative possibility that ARID1A might promote the sex-linked enrichment of H3.3 by influencing the expression of cognate chaperones like DAXX and HIRA during murine pachynema (*Rogers et al., 2004*; *van der Heijden et al., 2007*). We addressed this by monitoring DAXX and HIRA levels in response to the loss of ARID1A using IF. In the case of DAXX, we examined meiotic spreads to find no changes in either the overall levels or sex chromosome localization of DAXX in

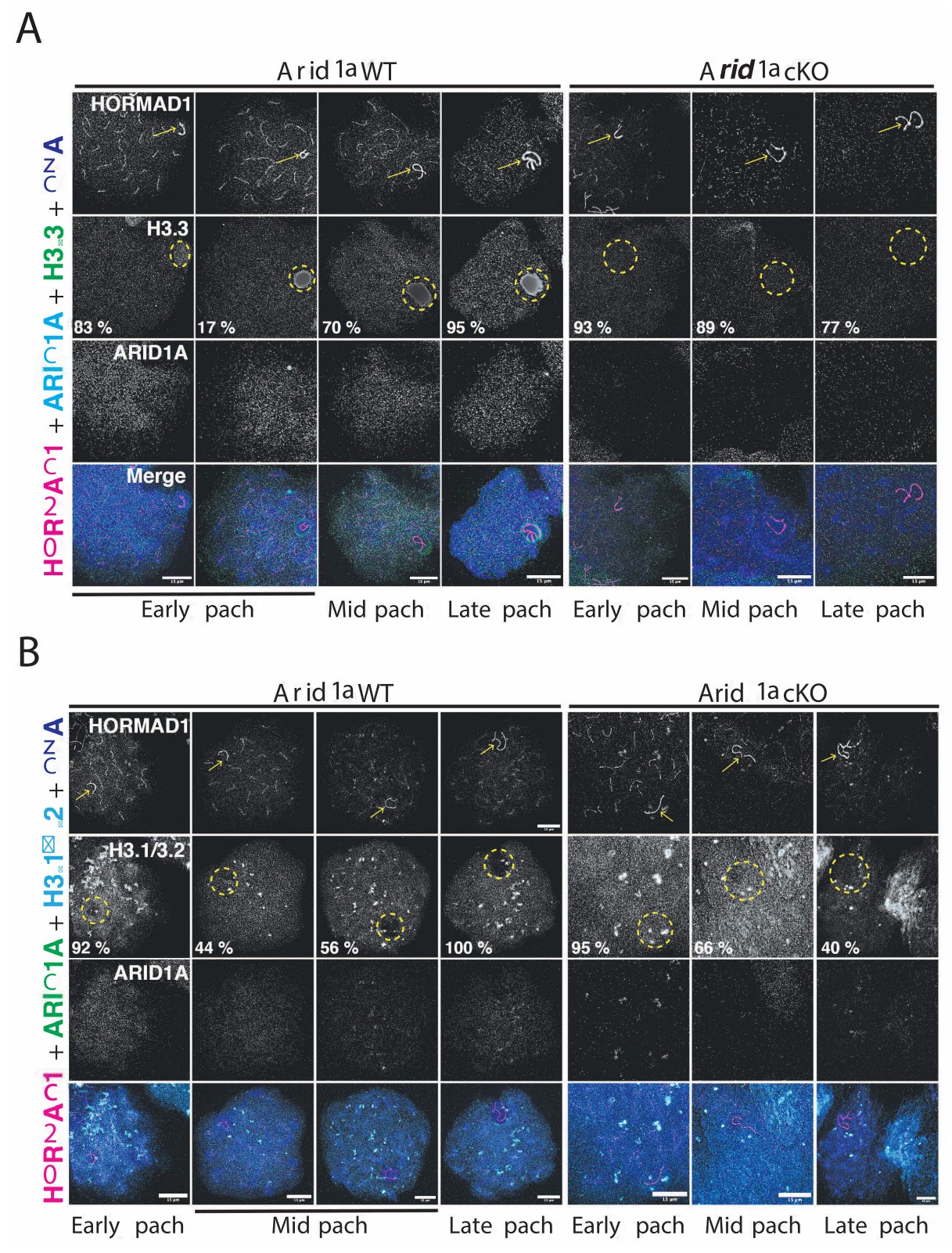

**Figure 5.** ARID1A influences the chromatin composition of the sex body. (**A–B**) *Arid1a* WT and *Arid1a* cKO pachytene spermatocytes immunolabelled for HORMAD1 (magenta), (**A**) ARID1A (cyan) and H3.3 (green), (**B**) ARID1A (green) and H3.1/3.2 (cyan). (**A–B**) Proportion (%) of *Arid1a* WT and *Arid1a* cKO early, mid, and late pachytene spermatocytes displaying distinct (**A**) H3.3, (**B**) H3.1/3.2 localization patterns with the sex body. For H3.3 immunostaining, the total number of *Arid1a* WT pachytene spermatocytes: early = 144, mid = 274, late = 95; *Arid1a* cKO pachytene spermatocytes:

*Figure 5 continued on next page*

*Figure 5 continued*

early = 43, mid = 86, late = 55, were scored, from three replicates each. For H3.1/3.2 immunostaining, the total number of *Arid1a* WT pachytene spermatocytes: early = 91, mid = 124, late = 45; *Arid1a* cKO pachytene spermatocytes: early = 22, mid = 116, late = 58, were scored, from three replicates each. DNA counterstained with DAPI (blue). The sex chromosomes (yellow arrow) and sex body (yellow dashed circle) are labeled. Scale bar:15 μm, magnification: ×100. Brightness of the whole image panel (Figure 5) was increased by compressing the dynamic range of the image panels using Adobe photoshop (0-1-255 to 0-1-150).

The online version of this article includes the following source data and figure supplement(s) for figure 5:

**Figure supplement 1.** ARID1A does not influence the expression or incorporation of H3.3.

**Figure supplement 1—source data 1.** Raw images of western blot for H3.3 and total histone (*Figure 5—figure supplement 1a*).

**Figure supplement 1—source data 2.** Complete and marked images of western blot for H3.3 and total histone (*Figure 5—figure supplement 1a*).

**Figure supplement 2.** ARID1A limits H3.3 occupancy at promoters.

**Figure supplement 2—source data 1.** MACS2 H3.3 peak calls from *Arid1a* WT and *Arid1a* cKO pachytene spermatocytes.

**Figure supplement 3.** Genome browser views of ARID1A governed H3.3 genomic associations.

response to the loss of ARID1A (*Figure 5—figure supplement 1B*). The same was also true of HIRA, whose levels and nuclear localization in mutants appeared comparable to that seen in escaper (from *Arid1a* cKO) and normal (*Arid1a* WT) spermatocytes (*Figure 5—figure supplement 1C*). Therefore, ARID1A impacts H3.3 accumulation on the sex chromosomes without affecting its expression during pachynema. Overall, our data suggest that ARID1A influences MSCI by regulating the composition of sex-linked chromatin.

## ARID1A prevents the promoter accumulation of H3.3

To define the changes in the sex-linked associations of H3.3 at a higher resolution, we performed CUT&RUN on pachytene spermatocytes isolated from *Arid1a* WT and *Arid1a* cKO testes. Despite the appearance of a robust H3.3 IF signal from the sex body during normal pachynema (*Figure 5A*), the Macs2 peak caller (*Zhang et al., 2008*) identified very few sex-linked peaks (n=224). These peaks were overwhelmingly associated with intergenic regions (*Figure 5—figure supplement 2A*). Comparatively, a dramatically higher number of peaks (n=12183) primarily associated with genic (promoter, intron, and exon) regions occurred in *Arid1a* cKO pachytene spermatocytes (*Figure 5—figure supplement 2A*). More interestingly, there appeared to be a shift from few but overwhelmingly sex-linked H3.3 peaks (n=224; 98%) to autosomal H3.3 peaks (n=11512; 94.5 %) in response to the loss of ARID1A (*Figure 5—figure supplement 2A*). Additionally, while the proportion of H3.3 sex-linked peaks in *Arid1a* cKO pachytene spermatocytes was few (n=671; 5.5%), they outnumbered their *Arid1a* WT counterparts by threefold. These data suggest that the loss of ARID1A strongly influences H3.3 genomic associations.

Due to the increased representation of promoter-associated peaks in response to the loss of ARID1A, we monitored H3.3 association with TSSs of genes differentially regulated by ARID1A. Consistent with the genomic annotation of H3.3 peaks, we could not detect H3.3 enrichment at TSSs associated with ARID1A-regulated sex-linked genes relative to IgG control. H3.3 signal spread gene-wide in *Arid1a* WT pachytene spermatocytes (*Figure 5—figure supplement 2B*, *Figure 5—figure supplement 3A-top panel*). In contrast, we observed a striking accumulation of H3.3 at the TSSs of sex-linked genes misexpressed without ARID1A (*Figure 5—figure supplement 2B*, *Figure 5—figure supplement 3A*). There is a transition from the gene-wide spreading of H3.3 to promoter-proximal enrichment in response to the loss of ARID1A. Concomitantly, X and Y-linked intergenic regions undergo a significant decrease in H3.3 occupancy upon ARID1A loss (*Figure 5—figure supplement 2A*). A similar loss of H3.3 occupancy occurred at a handful of autosomal intergenic peaks (*Figure 5—figure supplement 2A*, *Figure 5—figure supplement 3B* - top).

Unlike the sex-linked targets, TSSs of ARID1A-regulated autosomal genes displayed H3.3 enrichment relative to the IgG control (*Figure 5—figure supplement 2C*, *Figure 5—figure supplement 3B* - bottom). However, this TSS occupancy appeared enhanced in *Arid1a* cKO relative to *Arid1a* WT pachytene spermatocytes (*Figure 5—figure supplement 2C*, *Figure 5—figure supplement 3B*). Therefore, TSSs of both sex-linked and autosomal gene targets display concordant changes in H3.3 occupancy. Overall, ARID1A occupancy at TSSs (*Figure 2—figure supplement 5*) of differentially regulated genes prevents the accumulation of H3.3 at target promoters.

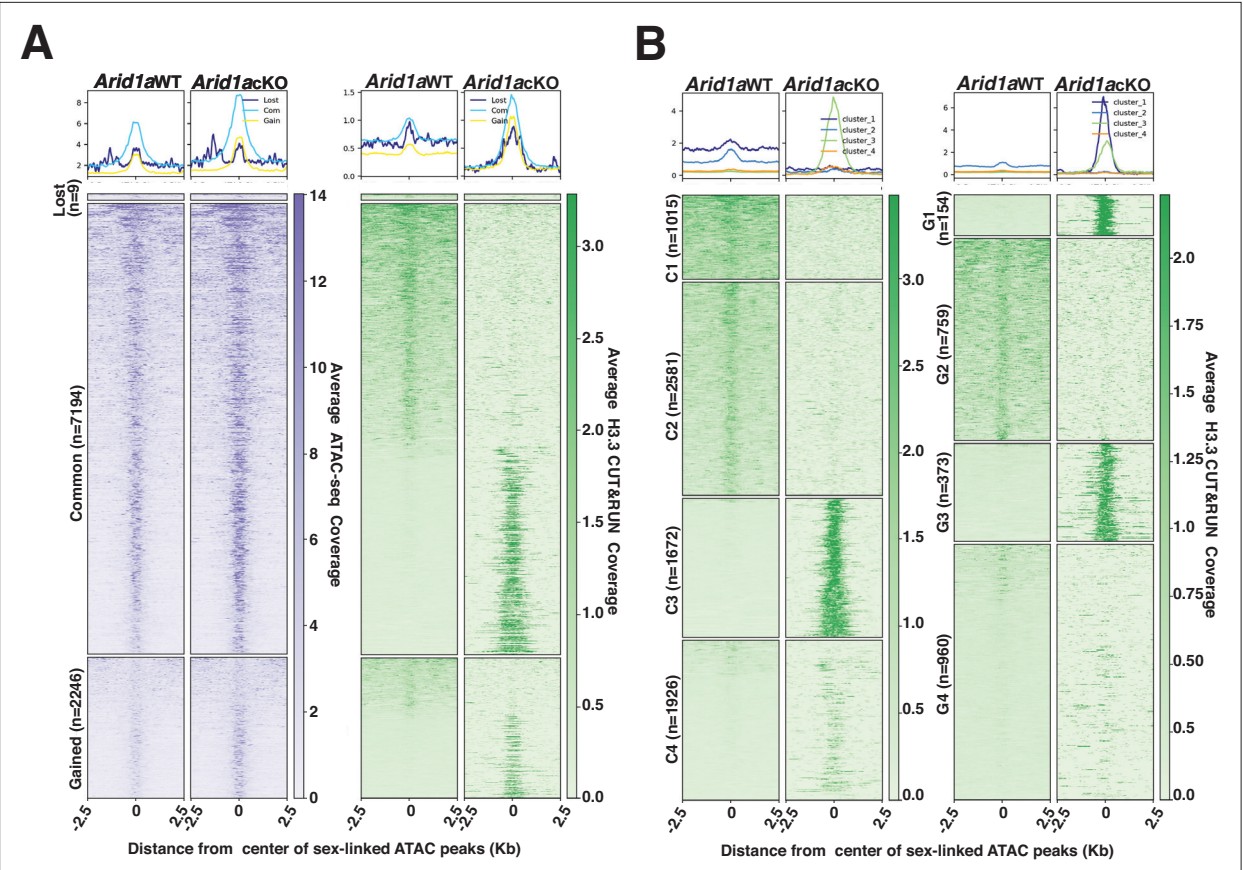

**Figure 6.** H3.3 displays differential occupancy across ARID1A-governed sex-linked open chromatin. (**A–B**) Heatmaps (bottom) and metaplots (top) displaying average (**A**) chromatin accessibility (purple heatmap) and H3.3 enrichment (green heatmap) associated with lost, common, and gained ATAC-seq peak calls (MACS2), (**B**) enrichment of H3.3 at k-means clusters associated with common (C1-C4, left) and gained (G1-G4, right) ATAC-seq peaks, in *Arid1a* WT and *Arid1a* cKO pachytene spermatocytes. (**A–B**) ATAC-seq and H3.3 coverage plotted over a 5 Kb window centered at ATAC-seq peaks (MACS2). Number of ATAC-seq peak calls (**n**) associated with each category is indicated.

The online version of this article includes the following figure supplement(s) for figure 6:

**Figure supplement 1.** H3.3 occupancy is antagonistic to DMC1 associations in non-homologous sex-linked regions.

## ARID1A restricts H3.3 occupancy to intergenic regions on the sex chromosomes

Our analysis of the genome-wide localization of H3.3 in response to the loss of ARID1A would argue that the abnormal increase in promoter-associated H3.3 results from its redistribution from high-affinity sites (peaks) that are predominantly sex-linked and intergenic during normal pachynema (*Figure 5—figure supplement 2A*, *Figure 5—figure supplement 3*). We focused our attention on these H3.3-occupied intergenic sex-linked sites. Although the Macs2 algorithm identified only 224 sex-linked intergenic peaks, we speculated that this approach filtered out regions enriched for H3.3 that failed to meet the peak calling threshold compounded by the pervasive spreading of H3.3 on the sex chromosomes (*Figure 5A*, *Figure 5—figure supplement 2B*, *Figure 5—figure supplement 3A*). We adopted an alternative strategy that involved identifying potential H3.3 bound sites by monitoring their occupancy at ARID1A-governed accessible chromatin (*Figure 4A*). We compared *Arid1a* WT and *Arid1a* cKO pachytene spermatocyte-associated ATAC-seq peaks to identify mutually exclusive and overlapping peaks using BEDTools (*Quinlan and Hall, 2010*). The resulting peaks cluster into lost (n=9), common (n=7194), or gained (n=2246) based on their absence, persistence, or appearance, respectively, in *Arid1a* cKO relative to *Arid1a* WT pachytene spermatocytes (*Figure 6A*). Next, we examined the ATAC-seq signal at these regions to find an increase in accessibility at gained peaks as expected and at common peaks in *Arid1a* cKO relative to *Arid1a* WT pachytene spermatocytes.

Despite being labeled as lost, these regions did not display the expected loss in ATAC-seq signal in *Arid1a* cKO relative to *Arid1a* WT pachytene spermatocytes (*Figure 6A*, left panel). The ATAC signal appears unchanged, highlighting the possibility that lost regions were labeled as such because they failed to satisfy Macs2 peak thresholds in *Arid1a* cKO pachytene spermatocytes, especially given the comparatively higher magnitude of ATAC signal surrounding the common and gained peaks (*Figure 6A*, left panel). We restricted our analysis to common and gained sites constituting 99% of sex-linked regions subject to ARID1A-regulated chromatin accessibility.

By plotting the average H3.3 coverage at sites associated with common and gained peaks, we detected striking differences in H3.3 occupancy. Briefly, both common and gained open chromatin regions consisted of loci differentiated by either a loss or robust increase in H3.3 occupancy in *Arid1a* cKO relative to *Arid1a* WT pachytene spermatocytes (*Figure 6A*, right panel). To gain further insight, we performed k-means clustering, based on H3.3 occupancy, to identify four clusters associated with either common (C1-C4) or gained (G1-G4) peaks (*Figure 6B*). The clusters associated with common ATAC-seq peaks displayed contrasting changes in H3.3 occupancy in response to the loss of ARID1A. C1 and C2 represented regions deficient for H3.3, while C3 and C4 represent regions that gained H3.3 binding in *Arid1a* cKO relative to *Arid1a* WT pachytene spermatocytes (*Figure 6B*, right panel). Similarly, clusters associated with gained ATAC-seq peaks, namely, G1 and G3, represented sites that gained H3.3 occupancy, while G2 identified sites that lost H3.3 occupancy. In contrast to clusters G1-G3, regions associated with G4 displayed an unremarkable association with H3.3 in response to the loss of ARID1A (*Figure 6B*, right panel). Genomic annotations of common and gained k-means clusters revealed that the sites that displayed a loss of H3.3 binding (represented by C1 +C2, G2) in response to the loss of ARID1A were intergenic (*Figure 6—figure supplement 1A*).

In contrast, those that gained significant H3.3 localization (represented by C3, C4, and G1 +G3) distributed differentially between intergenic and genic regions (*Figure 6—figure supplement 1A*). Therefore, these data support our idea that ARID1A restricts H3.3 occupancy primarily to intergenic sites on the sex chromosomes. Furthermore, the loss of ARID1A triggers an abnormal redistribution of H3.3 to genic regions. Therefore, the increased promoter occupancy of H3.3 is an indirect consequence of ARID1A loss.

## ARID1A-governed H3.3 localization influences the sex-linked association of DMC1

To gain further insight into the mechanisms regulating H3.3 associations on the sex chromosomes, we examined the regions associated with various k-means clusters to investigate the enrichment of relevant motifs. To our surprise, we detected an enrichment of the motif associated with the DSB hotspot specifier, PRDM9 (PR domain-containing protein 9), in clusters related to gained peaks, G1 and G3 (*Figure 6—figure supplement 1B*, top). Additionally, we observed an increased frequency of PRDM9 motif centered at gained ATAC-seq peaks associated with G1 and G3 (*Figure 6—figure supplement 1B*, middle).

Interestingly, regions associated with G1 and G3 are usually devoid of H3.3 (*Figure 6B*), implying a potential antagonism between H3.3 occupancy and the occurrence of meiotic DNA DSBs. Gained ATAC-seq peaks associated with G2, which usually displays H3.3 occupancy, appeared devoid of PRDM9 motif occurrences relative to G1 and G3 (*Figure 6—figure supplement 1B*, middle). Next, we determined the frequency of the PRDM9 motif at k-means clusters associated with common ATAC-seq peaks. These regions also displayed a dichotomous association with H3.3 (*Figure 6B*). C3 and C4, usually deficient in H3.3, showed an increased frequency of PRDM9 motifs relative to C2, which typically displays H3.3 binding (*Figure 6—figure supplement 1B*, bottom). These data illustrate an antagonistic relationship between H3.3 and meiotic DNA DSBs.

Since the PRDM9 motifs identified from SSDS (Single-Stranded DNA Seq) data were generated to map the genomic associations of DNA repair factor DMC1 (*Brick et al., 2012*), we monitored its enrichment at the various k-means clusters. We also monitored the association of markers of meiotic DNA DSBs, such as the PRDM9 catalyzed H3K4me3 at common and gained k-means clusters using published ChIP-seq data from pachytene spermatocytes (*Maezawa et al., 2018a*). Consistent with the motif analyses, DMC1 occupancy and H3K4me3 enrichment occurred at gained and common ATAC-seq peaks associated with k-means clusters, normally devoid of H3.3, namely, G1, G3, G4, C3, and C4 (*Figure 6—figure supplement 1C, D*).

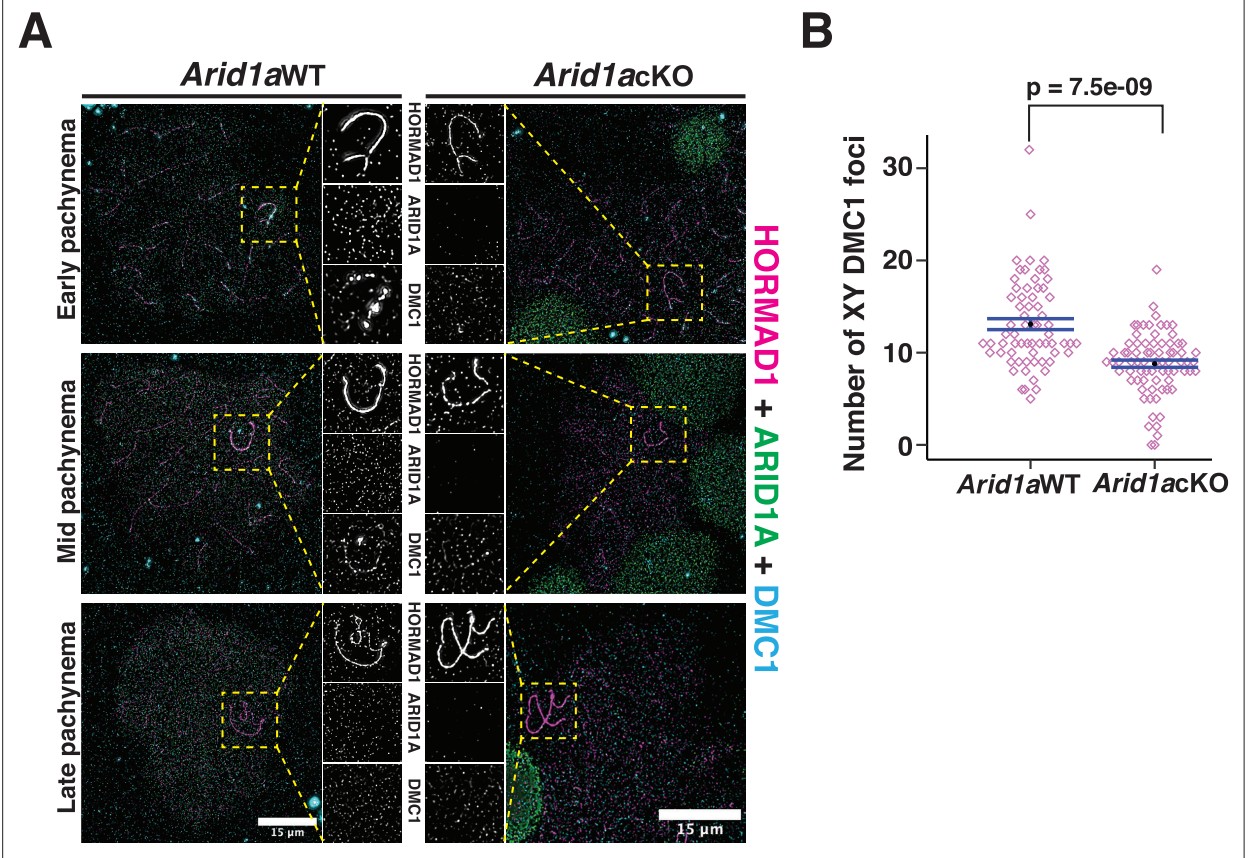

**Figure 7.** ARID1A influences the axial association of DMC1 with the XY during pachynema. (**A**) *Arid1a* WT and *Arid1a* cKO pachytene spermatocytes immunolabelled for HORMAD1 (magenta), ARID1A (green), DMC1 (cyan), and counterstained with DAPI (blue). Scale bar:15 μm, magnification: ×100. A magnified view of the sex chromosomes is indicated (yellow dashed square). (**B**) Dot plot displaying the number of sex-linked DMC1 foci (y-axis) quantified from *Arid1a* WT (n=66) and *Arid1a* cKO (ARID1A⁻, n=75) pachytene spermatocytes, obtained from three replicates each. Empty diamonds (magenta) represent independent data points. Significance determined using a two-tailed unpaired Student's t-test p values. Data expressed as mean (black dot) ± SEM.

The online version of this article includes the following source data and figure supplement(s) for figure 7:

**Source data 1.** Quantitation of DMCl Foci on the sex chromosomes.

**Figure supplement 1.** ARID1A does not affect the axial association of RAD51 with the XY during pachynema.

These data suggest that ARID1A might indirectly influence DNA DSB repair on the sex chromosomes by regulating the localization of H3.3. To test this conclusion, we monitored the association of DMC1 on the sex chromosomes in response to the loss of ARID1A by IF. Normally, DMC1 foci appear distributed along the non-homologous arms of the X and Y chromosomes from early to mid-pachynema, only disappearing by late pachynema (*Figure 7A*, left, panel insets; *Moens et al., 2002*). In contrast, in the absence of ARID1A, we observed a marked reduction in the number of DMC1 foci (*Figure 7A*, right, panel insets). Quantification of DMC1 foci in *Arid1a* cKO relative to Arid1a WT pachytene spermatocytes revealed a significant decrease in the sex-linked association of DMC1 in the absence of ARID1A (*Figure 7B*, *Figure 7—source data 1*). Along with DMC1, the mitotic DNA recombinase, RAD51, is also known to localize to the non-homologous arms of the sex chromosomes (*Moens et al., 1997*; *Moens et al., 2002*). Therefore, we determined whether ARID1A also influenced the sex-linked association of RAD51. First, we analyzed previously generated RAD51-SSDS data (*Hinch et al., 2020*) to monitor its enrichment across the various k-means clusters associated with common and gained ATAC-seq peaks. Unlike DMC1, we found no detectable levels of RAD51 at any k-means clusters (*Figure 7—figure supplement 1A*). Furthermore, the formation of RAD51 foci along the asynapsed axes of the sex chromosomes appeared unchanged in *Arid1a* cKO relative to *Arid1a* WT pachytene spermatocytes (*Figure 7—figure supplement 1B*). Therefore, ARID1A is necessary for

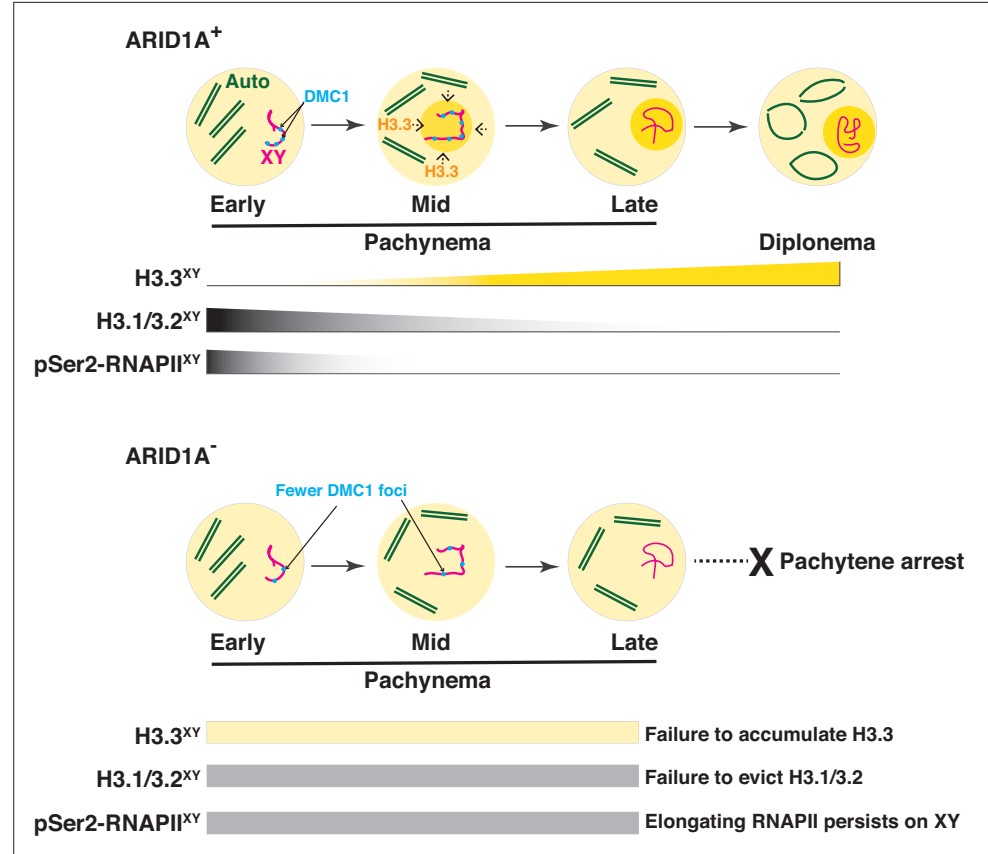

**Figure 8.** A model describing the role of ARID1A sex-linked chromatin regulation. During meiosis, the sex chromosomes undergo transcriptional repression at the onset of pachynema. A dramatic change in the composition of sex-linked chromatin accompanies this chromosome-wide repression. From mid to late pachynema, spermatocytes display a hyper-accumulation of the variant histone H3.3 (Ochre shading and gradient) on the sex chromosomes (magenta) relative to autosomes (green). Concomitantly, the canonical histones H3.1/3.2 levels and elongating pSer2-RNAPII complex (grey gradients) appear depleted from the sex body by late pachynema. The loss of ARID1A dramatically alters the chromatin composition of the sex body, which features low H3.3 (yellow bar) association at levels indistinguishable from autosomes throughout pachynema. Concomitantly, canonical H3.1/3.2 and pSer2-RNAPII levels (grey bar) on the sex body remain abnormally stable throughout pachynema. These sex-linked chromatin aberrations, along with persistent transcription owing to the association of pSer2-RNAPII with mutant sex body, fail meiotic sex chromosome inactivation (MSCI) and, consequently, pachytene arrest. This defect also coincides with an abnormal loss of DMC1 (blue foci) localization to the unpaired sex chromatids in response to the loss of ARID1A. Therefore, along with transcriptional repression, ARID1A-governed chromatin dynamics appear to influence DNA repair on the sex chromosomes.

the normal sex-linked association of DMC1 but not RAD51. More importantly, our data highlight a dual role for ARID1A in regulating sex chromosome repression and DNA repair (*Figure 8*).

## Discussion

We previously demonstrated a spermatogenic requirement for the mammalian SWI/SNF complex based on its role in coordinating germline transcription (*Menon et al., 2019*). Furthermore, we showed that the SWI/SNF PBAF subcomplex activates genes essential for reductional meiosis and gamete formation (*Menon et al., 2021*). These data highlight that distinct SWI/SNF subcomplexes govern specific meiotic transitions.

Our current study demonstrates a requirement for the ARID1A-associated BAF (BAF-A) subcomplex for MSCI. It does so by promoting the accumulation of the variant histone H3.3 on the sex chromosomes at the onset of mid-pachynema, coinciding with the eviction of canonical histones H3.1/3.2 and RNA polymerase II from the sex body (Model, *Figure 8*). In contrast, when depleted

of ARID1A, chromatin accessibility increased along with a deficiency of H3.3 and the continued presence of H3.1/3.2 and pSer2-RNAPII (an elongating form of RNA polymerase II) in the sex body. These data indicate that BAF-A helps define features distinguishing sex chromosomes from autosomes and that the absence of the BAF-A subcomplex during meiosis I results in the failure of MSCI during pachynema.

Although previous work described the association of H3.3 with the sex body (*van der Heijden et al., 2007*; *Yuen et al., 2014*) and its requirement for meiotic repression of sex-linked genes (*Fontaine et al., 2022*), our study revealed the need for BAF-A in the preferential localization of H3.3 to the sex body relative to autosomes. Interestingly, a similar role for BAF-A in promoting H3.3 incorporation occurs in human endometriotic cells (*Reske et al., 2022*). In this case, the genomic localization of H3.3 to super-enhancers required ARID1A. Our study describes the ARID1A-dependent localization of H3.3 at a chromosome-wide scale comprised of (i) local enrichment at intergenic loci and (ii) larger domains that resembled spreading much like that of *Xist* during female X chromosome inactivation (XCI) (*Chaumeil et al., 2006*). Interestingly, X chromosome inactivation (XCI) requires SWI/SNF subunits SMARCA4 and SMARCC1 in female mouse embryonic stem cells (mESC; *Keniry et al., 2022*). Here, SMARCC1 promotes XCI by increasing promoter accessibility of the inactive X chromosome (Xi). Although sex-linked chromatin accessibility is thought to increase during MACI (*Maezawa et al., 2018b*), our studies showed that BAF-A limits rather than enhances sex-linked chromatin accessibility at pachynema. These data highlight a mechanism that checks chromosome-wide relaxation during MSCI. The divergent outcomes of SWI/SNF-governed sex-linked chromatin accessibility during XCI and MSCI probably reflect noticeable sex-specific differences or arise from potentially perturbing distinct SWI/SNF subcomplexes.

Along with altered chromatin composition and structure, pachytene spermatocytes lacking ARID1A also displayed an abnormal association of pSer2-RNAPII (elongating form) with the sex body. Normally, total transcriptional output peaks towards the end of meiotic prophase-I, even with MSCI coinciding (*Alexander et al., 2023*; *Ernst et al., 2019*; *Monesi, 1964*). The dynamic switching from the paused (pSer5-RNAPII) to the elongating (pSer2-RNAPII) state of RNAPII drives this burst of transcription on autosomes (*Alexander et al., 2023*). The association of pSer2-RNAPII with the sex chromosomes without ARID1A renders them more autosome-like, underlying the defect in sex-linked gene repression. The BAF-A-governed H3.3 dynamics on the sex chromosomes may limit the switch from the paused to the elongating state of RNAPII. In a human cell culture model, ZMYND11, a known tumor suppressor and reader of Lysine 36 trimethylation on H3.3 (H3.3K36me3), suppresses RNAPII elongation (*Wen et al., 2014*). Whether similar H3.3 readers affect RNAPII pausing during meiosis in mice remains unknown.

The initiation and maintenance of MSCI are known to be regulated by DNA damage response (DDR) factors, ATR, and MDC1, respectively (*Ichijima et al., 2011*; *Royo et al., 2013*; *Turner et al., 2004*). However, ARID1A deficient pachytene spermatocytes appeared proficient in sex-linked DDR signaling, evidenced by the normal enrichment of ATR, γH2Ax, and MDC1 on the sex body. It is possible that BAF-A-mediated transcriptional repression of the sex chromosomes occurs independently of DDR signaling. However, given that the accumulation of ATR and MDC1 on the sex body precedes that of ARID1A, which only manifests late in diplonema, it is reasonable to hypothesize that DDR signaling might recruit BAF-A to the sex chromosomes.

ARID1A did not appear necessary for sex-linked DNA DSB formation. In contrast, the expected association of DMC1, the meiosis-specific DNA recombinase, with the asynapsed axes of the sex chromosomes do require ARID1A. Interestingly, sex-linked intergenic sites displaying an ARID1A-dependent enrichment of H3.3 appeared depleted for DMC1. Consistent with this antagonism, the loss of ARID1A resulted in the re-distribution of H3.3 to regions normally associated with DMC1. However, a subset of normal DMC1-associated regions (G4) displaying a distinct chromatin composition with little or no occupancy of H3.3 may undergo different dynamics of ARID1A regulation. Although only correlative, these data suggest that regions displaying a localized enrichment of H3.3 might be unfavorable for DNA repair. Determining the extent to which a deficiency in BAF-A activity might impede sex-linked DNA repair is challenging. RAD51 binding to the sex chromosomes remained stable without ARID1A, suggesting that some sex-linked DNA repair remains. Furthermore, RAD51 and DMC1 bind ssDNA at PRDM9/SPO11 designated DSB hotspots. However, these recombinases occupy unique domains. DMC1 localizes nearest the DSB breakpoint, promoting strand exchange,

whereas RAD51 is further away (*Hinch et al., 2020*). Our data suggest that loss of ARID1A decreases DMC1 foci on the XY chromosomes without affecting RAD51. These findings indicate that BAF-A may play a role in the loading and/or retention of DMC1 to the XY chromosomes. Additionally, monitoring the formation of DMC1 and RAD51 foci early in meiosis I (Leptonema-Zygonema) in response to ARID1A loss would be interesting. However, this analysis would require a conditional mutation of *Arid1a* using a different *Cre* line. Nonetheless, our studies highlight a model in which BAF-A promotes meiotic progression by facilitating MSCI and DNA repair on the sex chromosomes.

# Materials and methods
.

## Key resources table

| Reagent type (species) or resource | Designation | Source or reference | Identifiers | Additional information |
|---|---|---|---|---|
| Strain (mouse) | *Arid1a*<sup>tm1Mag</sup>/Mmnc RRID:MMRRC_041418-UNC | Nat Commun. 2015 Jan 27;6:6118. (PMID:25625625) | MMRRC Stock number: 041418-UNC | Genotyping primers: *Arid1a*<sup>fl/+</sup> alleles - (F)- 5´ - CTAGGTGGAAGGTAGCTGACTGA –3´; (R) 5´- TACACGGAGTCAGGCTGAGC –3´ (PCR product sizes- *fl*: 300 bp; +: 200 bp) |
| Strain (mouse) | *Stra8-Cre* RRID;IMSR_JAX:008208 | Genesis. 2008 Dec;46(12):738–42 PMID:18850594 | JAX Stock # 008208 | *Stra8-Cre* - (F) 5´-GTGCAAGCTGAACAACAGGA -3´, (R) 5´-AGGGACACAGCATTGGAGTC-3´ (PCR product size- *Cre*: 150 bp) |
| Gene (mouse) | *Arid1a* cDNA | This Report | | primers: (F)- 5´ - TCCAGTAAGGGAGGGCAAGA AGAT –3´; (R) 5´ - GTAGTTGGCGTTGGGCAAGG CATTA –3´ (PCR product sizes- *fl*: 612 bp; Δ: 281 bp) |
| Antibody | anti- ARID1A RRID:AB_1078205 (Rabbit polyclonal) | Millipore Sigma | Cat # HPA005456 | IF (1:500) CUT&RUN (1:25) |
| Antibody | anti- ARID1A (clone D2A8U) RRID:AB_2637010 (Rabbit monoclonal) | Cell signaling technology | Cat # 12354 | IF (1:500) CUT&RUN (1:25) WB (1:1000) |
| Antibody | anti- SYCP3 (Cor10G1f1/7) RRID:AB_10678841 (Mouse monoclonal) | Abcam | Cat # ab97672 | IF (1:500) |
| Antibody | anti-phospho-Histone H2A.X, (Ser139) (clone JBW301) RRID:AB_309864 (Mouse monoclonal) | Millipore Sigma | Cat # 05–636 | IF (1:1000) |
| Antibody | anti-ATR (N-19) RRID:AB_630893 (Goat polyclonal) | Santa Cruz Biotech | Cat # sc-1887 | IF (1:50) |
| Antibody | anti-MDC1 RRID:AB_323725 (Sheep polyclonal) | AbD Serotec | AHP799 | IF (1:500) |
| Antibody | anti-RNA pol II CTD phospho Ser-2 (clone 3E10) RRID:AB_2687450 (Rat monoclonal) | Active Motif | Cat # 61084 | IF (1:500) |
| Antibody | anti- HORMAD1 (Guinea Pig polyclonal) | Gift from Dr. Atilla Töth, TU Dresden; *Wojtasz et al., 2009* | PMID:19851446 | IF (1:600) |
| Antibody | anti- H3F3B (Clone 2D7-H1) RRID:AB_425473 (Mouse monoclonal) | Abnova | Cat # H00003021-M01 | IF (1:200) CUT&RUN (1:25) |
| Antibody | anti- Histone H3.3 (Clone RM190) RRID:AB_2716425 (Rabbit monoclonal) | RevMAb Biosciences | Cat # 31-1058-00 | WB (1:1000) |

*Continued on next page*

*Continued*

| Reagent type (species) or resource | Designation | Source or reference | Identifiers | Additional information |
|---|---|---|---|---|
| Antibody | anti- Histone H3.1/3.2 (clone 1DA4F2) RRID:AB_2793710 (Mouse monoclonal) | Active Motif | Cat # 61630 | IF (1:500) |
| Antibody | anti-DAXX RRID:AB_2618529 (mouse monoclonal) | Developmental studies hybridoma bank | Cat # PCRP-DAXX-2B3 | IF (1:10) |
| Antibody | anti-HIRA (clone WC119.2H11) RRID:AB_2793256 (Mouse monoclonal) | Active Motif | Cat # 39558 | IF (1:100) |
| Antibody | anti- DMC1 (clone 2D5C9) RRID:AB_2882472 (Mouse monoclonal) | Proteintech | Cat # 67176–1-Ig | IF (1:200) |
| Antibody | anti- RAD51 RRID:AB_2665405 (Rabbit polyclonal) | Abcam | Cat # ab176458 | IF (1:500) |
| Antibody | anti-Nucleolin RRID:AB_533463 (Rabbit polyclonal) | Bethyl | Cat # A300-711A | WB (1:2000) |
| Antibody | anti-mouse IgG, Alexa fluor 488 RRID:AB_2534088 (Goat polyclonal) | Thermo Fisher | Cat # A-11029 | IF (1:500) |
| Antibody | anti-rabbit IgG, Alexa fluor 568 RRID:AB_10563566 (Goat polyclonal) | Thermo Fisher | Cat # A-11036 | IF (1:500) |
| Antibody | anti-goat IgG, Alexa fluor 633 RRID:AB_2535739 (Donkey polyclonal) | Thermo Fisher | Cat # A-21082 | IF (1:500) |
| Antibody | anti-sheep IgG, Alexa fluor 594 RRID:AB_2534083 (Donkey polyclonal) | Thermo Fisher | Cat # A-11016 | IF (1:500) |
| Antibody | anti-guinea pig IgG, Alexa fluor 568 RRID:AB_2534119 (Goat polyclonal) | Thermo Fisher | Cat # A-11075 | IF (1:500) |
| Antibody | anti- rat IgG, Alexa fluor 488 RRID:AB_2534074 (Goat polyclonal) | Thermo Fisher | Cat # A-11006 | IF (1:500) |
| Antibody | anti-mouse IgG1, Alexa fluor 647 RRID:AB_2535809 (Goat polyclonal) | Thermo Fisher | Cat # A-21240 | IF (1:500) |
| Antibody | anti- rabbit IgG, Alexa fluor 488 RRID:AB_143165 (Goat polyclonal) | Thermo Fisher | Cat # A-11008 | IF (1:500) |
| Antibody | anti-mouse IgG2b, Alexa fluor 647 RRID:AB_2535811 (Goat polyclonal) | Thermo Fisher | Cat # A-21242 | IF (1:500) |
| Antibody | anti- rabbit IgG, Alexa fluor 647 RRID:AB_2535813 (Goat polyclonal) | Thermo Fisher | Cat # A-21245 | IF (1:500) |

*Continued on next page*

Continued

| Reagent type (species) or resource | Designation | Source or reference | Identifiers | Additional information |
|---|---|---|---|---|
| Antibody | anti-mouse IgG2a, Alexa fluor 647<br>RRID:AB_2535810<br>(Goat polyclonal) | Thermo Fisher | Cat #<br>A-21241 | IF (1:500) |
| Antibody | IRdye 800CW anti-rabbit<br>RRID:AB_2651127<br>(Goat polyclonal) | LI-COR | Cat #<br>925–32211 | WB (1:10,000) |
| Antibody | AffiniPure anti-Mouse IgG (H+L)<br>RRID:AB_2340033<br>(Rabbit polclonal) | JacksonImmunoResearch | Cat #<br>315-005-003 | CUT&RUN (1:100) |
| Antibody | anti-Rabbit IgG (H+L)<br>RRID:AB_11024108<br>(Guinea pig polyclonal) | Novus biologicals | Cat #<br>NBP1-72763 | CUT&RUN (1:100) |

## Generation of *Arid1a* conditional mutant mice

Generation of the *Arid1a^{tm1Mag}*/Mmnc mutant allele was previously described (*Chandler et al., 2015*). The mice are available through the regional mutant mouse resource and research center (MMRRC; Stock number: 041418-UNC; https://www.mmrrc.org). *Arid1a* floxed mice (*Arid1a^{tm1Mag}*/*Mmnc*) exist on a CD1 genetic background. Crosses with testes-specific *Stra8-Cre* mice (expressed in P3 spermatogonia) (*Sadate-Ngatchou et al., 2008*) resulted in conditional knockouts. *Arid1a^{fl/fl}*; Stra8- Cre Tg/0 females were crossed to either *Arid1a^{fl/+}* or *Arid1a^{fl/fl}* males to obtain *Arid1a^{fl/fl}*; Stra8-Cre Tg/0 (*Arid1a* cKO) and either *Arid1a^{fl/fl}* or *Arid1a^{fl/+}* (*Arid1a* WT) males. Haploinsufficiency associated with *Arid1a* negated the possibility of transmitting the Cre through the paternal germline. Genotyping primers: *Arid1a^{fl/+}* alleles - (F)- 5ʹ - CTAGGTGGAAGGTAGCTGACTGA –3ʹ; (R) 5ʹ - TACACGGAGTCA GGCTGAGC –3ʹ (PCR product sizes- *fl*: 300 bp; +: 200 bp), and *Stra8-Cre* - (F) 5ʹ -GTGCAAGCTGAA CAACAGGA-3ʹ, (R) 5ʹ -AGGGACACAGCATTGGAGTC-3ʹ (PCR product size- *Cre*: 150 bp). Mice were housed with a 12 hr light cycle (temperature - 20-24° C, humidity - 30–70%). All animal work followed approved UNC Chapel Hill IACUC protocols.

## Histology

*Arid1a* WT and *Arid1a* cKO testes and cauda epidymites were fixed overnight in Bouins solution (Fisher Scientific Ricca chemical; 11–201) at 4° C, followed by dehydration in ethanol series (50%, 70%, and 100%) before embedding in paraffin. The animal histopathology core prepared stained tissue sections (hematoxylin and eosin for cauda epidymites; Periodic acid-Schiff for testes). Staining and morphology determined seminiferous tubule staging (*Ahmed and Rooij, 2009*; *Meistrich and Hess, 2013*). A summary of staging is provided (*Figure 2—figure supplement 2—source data 1*).

## Immunofluorescence staining of testis cryosection and spermatocyte spreads

Testes cryosections and spermatocyte spreads from juvenile (P18, P19, and P26) and adult *Arid1a* WT and *Arid1a* cKO mice were examined with indirect immunofluorescence (IF). 8–10 μm thick cryosections of testes were prepared using published methods (*Menon et al., 2021*). Whole testes were fixed in 10% neutral buffered formalin (NBF) at 4°C for 20 min. They were cut in half, followed by 40 min of incubation in 10% NBF. Fixed tissues were washed three times (10 min per wash) in room temperature (RT) phosphate-buffered saline (PBS) pH7.4. The tissues were then treated with a series of sucrose washes - 10% (30 min), 20% (30 min), and 30% (1 hr). Tissues were then incubated at 4°C overnight in a 1:1 ratio of a solution of 30% sucrose/optimum cutting temperature (OCT, Tissue-Tek; 62550–01), followed by embedding in OCT, sectioning, and storage at –80° C. Before immunostaining, cryosections were thawed on a heating block at 42° C, rehydrated for 10 min in PBS, and incubated in 10 mM citric acid buffer (pH6.0) for 10 min at 80–90°C (antigen retrieval), during which addition of fresh citrate buffer (80–90°C) occurred every 2 min, and then allowed to cool for 20 min. For immunostaining, cryosections were incubated overnight in a humidified chamber with primary at 4°C and then followed by a secondary antibody incubation at RT for 1 hr (see Key Resource Table for antibody

source). Sera were diluted with antibody dilution buffer in PBS (ADB: 3% bovine serum albumin; 10% goat serum; 0.5% Triton-X 100). Two identical blocking steps preceded antibody incubations. These steps involved sequential incubations in (i) PBS, (ii) PBS/0.1% Triton-X 100, and (iii) blocking solution (1:10 dilution of ADB in PBS) for 10 min each at RT. Immunostained slides were washed twice in Kodak Photo-Flo 200 (PBS/0.32%), counterstained with DAPI, washed again in PBS/0.32% Photo-Flo 200, and then mounted in Prolong Gold antifade medium (P-36931; Life Technologies).

Spermatocyte spreads were prepared as previously described (*Gray et al., 2020*) with minor modifications. Briefly, seminiferous tubules were incubated in hypotonic buffer (30 mM Tris pH8.2, 50 mM Sucrose, 17 mM Trisodium dihydrate, 5 mM EDTA, 0.5 mM DTT, 0.1 mM PMSF) for 10–20 min on ice. Tubules were minced in 100 mM sucrose (200 µL/testis) until the solution turned turbid. The resulting cell suspension was isolated, avoiding tissue debris, and held on ice. Cells were fixed and spread by dropping 20 µL of cell suspension directly onto glass slides coated with paraformaldehyde solution (1% Paraformaldehyde; 0.15% Triton X-100; 625 nM Sodium Borate, pH 9.2). Uniform spreading was ensured by tilting the slides along the near and opposite edges. Next, slides were incubated in a humidified chamber for 1 hr, following which they quickly air-dried and then washed two times in PBS/0.32% Photo-Flo 200 and once in $H_2O$/0.32% Photo-Flo 200. Air-dried slides were immediately processed for immunostaining as described above for cryosections or stored at -80°C. Immunostained spreads were washed 2 x with PBS/0.32% Photo-Flo 200, counterstained with DAPI, and washed again with Photo-Flo 200. Stained spreads were mounted in Prolong Gold antifade medium. The list of antibodies for IF is provided in the Key Resource Table. The Leica Dmi8 fluorescent microscope was used to capture images. Z-stacks were deconvoluted using Leica Dmi8 image software (Huygens essentials version 20.04). Some whole immunofluorescence image panels with low visibility (see figure legends 3,5 and figure Supplement legends 4,6,7) were enhanced to increase brightness.

Fluorescence intensity measurements and object (foci) counting were performed with Fiji (*Schindelin et al., 2012*). RNAPII (pSer2) IF signal intensities were measured using the method described previously (*Menon et al., 2021*). Briefly, pixel intensities were recorded from regions of interest (ROI), which include the sex chromosomes (sex body) and an area lacking cells (background fluorescence). Corrected total RNAPII-pSer2 fluorescence (CTRF) was calculated using the method previously described (here), where CTRF = Integrated density associated with sex body – (Area of sex body x Mean fluorescence of background).

For counting axial DMC1 foci, images were cropped to an area encompassing the sex chromosomes. Cropped images were converted to 8-bit grayscale, and the DMC1 fluorescent channel selected for further processing. An ROI was set, manual thresholding was performed to select DMC foci, and noise (speckles) removed using the Fiji median filter tool with a radius set to 2.0–4.0. Finally, DMC1 was enumerated with the count particles tool. Statistical significance was calculated using an unpaired student's t-test. Metadata associated with the quantification of RNAPII (pSer2) fluorescence and DMC1 foci are provided (*Figure 7—source data 1*).

## Preparation of acid-extracted histones and western blotting

Histones from P19 *Arid1a* WT and *Arid1a* cKO spermatogenic cells were acid-extracted using published methods (*Shechter et al., 2007*). Seminiferous tubules were digested in 1 mL of Enriched Krebs-Ringer Bicarbonate (EKRB) Medium (EKRB salts - 1.2 mM $KH_2PO_4$, 1.3 mM $CaCl_2$, 119.4 mM NaCl, 4.8 mM KCl, 1.2 mM $MgSO_4$, 11.1 mM Dextrose; 25.2 mM $NaHCO_3$; 1 X essential amino acids -Invitrogen, 11130051; 1 X non-essential amino acids-Gibco,11140050) supplemented with 0.5 mg/mL collagenase (10 min, rotated at 32 °C). Digested tubules sedimented by gravity for 5 min at RT. The supernatant was removed, tubules were resuspended in 1 mL EKRB medium, and then settled for 5 min at RT. The sedimented tubules were transferred to 1 mL EKRB medium supplemented with 0.025% trypsin and 4 µg/mL DNase-I. The suspension was incubated for 10 min on a rotator at 32°C. Digestion was halted with 10% Fetal bovine serum (FBS), and single-cell suspensions were prepared by pipetting the slurry and then filtering it sequentially through 70 µm and 40 µm filters. Cells were pelleted at 600 × *g* for 5 min, washed once in PBS, and processed for histone extraction. Briefly, cell pellets were resuspended in 1 mL hypotonic lysis buffer (10 mM Tris-Cl pH8.0, 1 mM KCl, 1.5 mM $MgCl_2$, 1 mM DTT) and incubated on a rotator at 4°C for 30 min. Nuclei were pelleted at 10,000 × *g* for 10 min at 4° C. Nuclear pellets were resuspended in 400 µL 0.2 N HCl and incubated for 1 hr at 4 °C, followed by centrifugation at 14,000 × *g* for 10 min at 4°C to isolate supernatants containing histones

that were precipitated with Trichloroacetic acid (TCA), and then added to the nuclear suspension at a final concentration of 33% and incubated on ice for 30 min. Precipitated histones were pelleted at 14,000 × g for 10 min at 4°C and washed two times in 1 mL ice-cold acetone. Finally, washed histones were air-dried at RT for 20 min and resuspended in 100 μL ddH$_2$O. Histones were separated on a 15% SDS-polyacrylamide gel for Western blots and transferred to a PVDF (Polyvinylidene Difluoride) membrane using a semi-dry transfer apparatus (Bio-Rad). Sample loading was assessed by REVERT total protein stain (LI-COR, 926–11010). Blots were scanned on a LI-COR Odyssey CLx imager and viewed using Image Studio Version 5.2.5. Antibodies and their corresponding dilutions are listed in the Key Resource Table.

## Isolation of pachytene spermatocytes by the Sta-Put method of unit gravity and western blotting

Pachytene spermatocytes were purified from *Arid1a* WT and *Arid1a* cKO adult testis by the unit gravity sedimentation procedure using a Sta-Put apparatus (**Bellvé, 1993**; **O'Brien, 1993**). Testis from at least 12 adult males (>P60) were dissected into EKRB, decapsulated, and digested in 0.5 mg/mL collagenase for 15 min at 32°C. The resulting tubules were gently washed with EKRB and digested with 0.25 mg/ml trypsin supplemented with 4 μg/mL DNase for 15 min at 32°C. Following trypsin incubation, additional DNase (4 μg/mL) was added to reduce viscosity further. Samples were triturated for 3 min with a plastic transfer pipet to fragment flagella, break intercellular bridges between germ cells, and further disperse cells into suspension. Trypsin digestion was stopped by adding 0.25 mg/mL trypsin inhibitor (Sigma/EPRO, T9003), and cell aggregates were removed by filtration using 40 μm cell strainers. Cells from the flow-through were pelleted at 500 × g for 5 min at 4°C, resuspended in EKRB +0.5% BSA, and adjusted to a 3.33x10$^8$ cells/mL density. 10$^9$ cells from this single-cell suspension were loaded onto a 2–4% linear BSA gradient and allowed to sediment in the Sta-Put apparatus at 4°C. After 2 hr and 40 min of undisturbed sedimentation, 10 ml fractions were collected at 45 s per tube. Fractions were pelleted at 500 × g for 5 min at 4 °C and resuspended in 1 ml EKRB +0.5% BSA. Fractions containing >80% pachytene spermatocytes were identified by light microscopy, pooled, and stored in cryoprotective media (DMEM supplemented with 10%FBS and 20%DMSO) at –80°C pending further analysis.

Pachytene spermatocytes obtained were resuspended in RIPA buffer (50 mM Tris-HCl pH 8.0, 150 mM NaCl, 1% NP-40, 0.5% Sodium deoxycholate, 0.1% SDS, 5 mM NaF, 1 mM Na$_3$VO$_4$, 1 mM Phenylmethylsulfonyl fluoride (PMSF), 1 X Protease inhibitor cocktail (Sigma)) at 4 °C and sonicated for 15 min followed by centrifugation at 12000 x g for 10 min at 4°C. Western blotting was performed by separating samples in 5% SDS-polyacrylamide gel and transferring to PVDF (polyvinylidene difluoride) membranes (Bio-Rad) overnight. Blots were scanned on a LI-COR Odyssey CLx imager. Image Studio Version 5.2.5 was used to view blots and quantify band intensity. Antibodies and their corresponding dilutions are listed in the Key Resource Table.

## Flow cytometry

To analyze isolated spermatocytes, the cells were resuspended in Hank's balanced salt solution (HBSS) (10$^6$ cells/ml) at 4 °C and stained with Live-or-Dye 615/740 fixable viability stain (Biotium) for 30 min. After washing the cells three times with HBSS, they were fixed for 1 hr and permeabilized using eBioscience Foxp3 /Transcription Factor Staining Buffer Set (ThermoFisher). DNA was stained using Vybrant DyeCycle Violet Stain (Thermo Fisher; 1:1000) and analyzed by flow cytometry using LSRFortessa (BD Biosciences). FCS files were obtained and analyzed using FlowJo software.

## RNA extraction and RT-PCR

Total RNA was isolated in TRIzol reagent (Invitrogen, 15596026) from Sta-Put purified fractions enriched for pachytene spermatocytes and round spermatids. The extracted RNA was purified using the Direct-zol RNA kit (Zymo, R2050). cDNA was synthesized using a random primer mix (New England Biolabs, S1330S) and ProtoScript II reverse transcriptase (New England Biolabs, M0368L). RT-PCR was performed on 1:10 cDNA dilutions using Sso Fast EvaGreen supermix (Bio-Rad, 172–5280) on a thermocycler (Bio-Rad, C1000). Amplicons associated with either floxed or excised *Arid1a* cDNA's were detected using primers: (F)- 5' - TCCAGTAAGGGAGGGCAAGAAGAT –3'; (R) 5' - GTAGTTGGCGTT GGGCAAGGCATTA –3' (PCR product sizes- *fl*: 612 bp; Δ: 281 bp).

## RNA-seq

RNA was extracted and purified in quadruplicate from frozen pellets of *Arid1a* WT and *Arid1a* cKO pachytene spermatocytes obtained by Salmony Sta-Put gravity sedimentation using the abovementioned methods. All samples displayed RNA integrity number (RIN) >7.0 as determined by Agilent Tapestation (High sensitivity RNA screen tape). Sequencing libraries were prepared from 1 µg of total RNA using a Kapa mRNA HyperPrep kit (Roche, KK8580) per the manufacturer's instruction. Libraries were quantified using Qubit dsDNA Hs assay kit (Invitrogen, Q32854) pooled at equimolar concentrations, and sequenced on a single lane of the Illumina NovaSeq 6000 S Prime flow cell (50 bp reads, paired-end).

## RNA-seq data analysis

Gene expression was quantified using Salmon (*Patro et al., 2017*). Transcript counts at the gene level were summarized using tximport (*Soneson et al., 2016*) and then imported to perform a differential gene expression analysis using DESeq2 (*Love et al., 2014*). The mouse (mm10) gene/transcript annotations were retrieved using AnotationDbi (*Pàges et al., 2021*) and *TxDb.Mus musculus.UCSC.mm10. known gene* (*Bioconductor Core Team, 2019*) R packages from Bioconductor. Low-count genes (<10 reads) were pre-filtered, and significant differences in counts were called at a false discovery rate (FDR)≤0.05. *Figure 2—source data 2* lists differentially expressed genes.

## CUT&RUN (cleavage under targets and release using nuclease)

CUT&RUN assays were performed on cryopreserved *Arid1a* WT and *Arid1a* cKO pachytene spermatocytes (500,000 cells/genotype) obtained by Sta-Put, using a slightly modified version of previously described methods (*Meers et al., 2019*; *Skene and Henikoff, 2017*). For ARID1A, CUT&RUN was performed in triplicates per genotype with two different antibodies. H3.3 CUT&RUN was performed in duplicates on *Arid1a* WT and triplicates on *Arid1a* cKO samples.

Aliquots of cryopreserved *Arid1a* WT and *Arid1a* cKO pachytene spermatocytes were quickly thawed in a water bath at RT, pelleted at 600 × *g* for 5 min, and washed [wash buffer: 20 mM HEPES(Na) pH7.5, 150 mM NaCl, 0.5 mM spermidine, EDTA-free protease inhibitor] three times. Cells were gently resuspended in wash buffer and bound to Concanavalin-A-coated magnetic beads (20 µL beads/500,000 cells) by mixing on a rotator for 15 min. The bead slurry was separated using a magnet followed by permeabilization in 50 µL antibody buffer (wash buffer +0.05% Digitonin +2 mM EDTA) per sample in 0.2 mL 8-strip PCR tubes. Primary antibody was added to the samples and incubated on a nutator at 4°C overnight. Cells were then separated using a magnet, washed two times in 200 µL Dig-wash buffer (wash buffer +0.05% Digitonin), and incubated with a secondary antibody prepared in 50 µL of Dig-wash buffer/sample for 30 min on a nutator at 4°C. After antibody incubations, samples were washed twice in 200 µL Dig-wash buffer and then incubated on a nutator at 4°C with 50 µL Dig-wash buffer containing 1000 ng/mL Protein A/G Micrococcal Nuclease (pA/G-MNase) for 1 hr. Following this, samples were washed twice in 200 µL Dig-wash buffer, resuspended in 50 µL Dig-wash buffer, cooled down to 0°C, and supplemented with $CaCl_2$ at a final concentration of 2 mM to activate MNase. Samples were digested at 0 °C for 40 min. The resulting chromatin fragments were extracted from the bead slurry by incubating in 2 X STOP buffer (340 mM NaCl, 20 mM EDTA, 4 mM EGTA, 0.05% Digitonin, 100 µg/mL RNase A) at 37°C for 30 min. DNA was purified using ChIP DNA clean and concentrator kit (Zymo, D5205). Libraries were prepared using the Kapa Hyperprep kit (Roche, KK8504), examined for size, and quantified on an Agilent 2100 bioanalyzer using the high-sensitivity DNA kit (Agilent, 5067–4626). The libraries were pooled in equimolar amounts and sequenced on a single lane of the Illumina NovaSeq 6000 S Prime flow cell (50 bp reads, paired-end). Antibody details are listed in the Key Resource Table. The pA/G-MNase (Addgene ID: 123461) used for the CUT&RUN experiments was purified at UNC-Chapel Hill's Protein Expression and Purification core.

## CUT&RUN data analysis

CUT&RUN data analyses were performed as previously described (*Menon et al., 2021*). Briefly, fastq files were analyzed with fastqc (version 0.11.9), then processed with TrimGalore (version 0.6.2) keeping the `--trim-n` option. Trimmed reads were aligned to mm10 (mouse) reference genome using bowtie2 (*Langmead and Salzberg, 2012*) parameters: bowtie2 `--very-sensitive-local --no-mixed --no-unal --dovetail --no-discordant`. Alignments were outputted

in BAM format using samtools (*Li et al., 2009*) and filtered for PCR duplicates using Picard tools, MarkDuplicates (https://broadinstitute.github.io/picard/). Deduplicated BAM files were used to generate bigWig files filtered for ENCODE mm10 blocked regions (*Amemiya et al., 2019*) with DeepTools, bamCoverage (*Ramírez et al., 2016*) with the following options: -bs 30 `--smooth-Length` 60 -bl `--scaleFactor`. The option `--scaleFactor` was used to normalize for composition bias between CUT&RUN libraries generated from *Arid1a* WT and *Arid1a* cKO pachytene spermatocytes.

csaw (*Lun and Smyth, 2016*) derived normalization factors (nf) were used to calculate the effective library size (library size X nf). The inverse of the effective library size per million was supplied to --scaleFactor to generate normalized bigWig files. Peak calling was performed with MACS2 (version 2.1.2) with options: -f BAMPE `--broad --broad-cutoff 0.05 --keep`-dup all, on filtered BAM files associated with ARID1A and H3.3 relative to an IgG control previously generated from pachytene enriched populations obtained from P18 testes (*Chakraborty and Magnuson, 2022*). Peak overlaps across replicates were determined using bedtools intersect (*Quinlan and Hall, 2010*). Peak calls are provided in *Figure 5—figure supplement 2—source data 1*. Peaks were annotated using HOMER (version 4.9.1), annotatePeaks.pl. Replicate bigWig files were averaged using WiggleTools (*Zerbino et al., 2014*). Averaged bigWig files generated heatmaps with DeepTools, computeMatrix, and plotHeatmap. Data was visualized on Integrative Genomics Viewer (*Robinson et al., 2011*).

## ATAC-seq

Cryopreserved aliquots of *Arid1a* WT and *Arid1a* cKO pachytene spermatocytes were processed in quadruplicate for ATAC-seq using the standard Omni-ATAC protocol (*Corces et al., 2017*) with minor changes. These involved increasing the cell number to 100,000 per replicate, using Diagenode tagmentase (loaded Tn5 transposase, C01070012) and Diagenode 2 X tagmentation buffer (C01019043). Libraries were quantified using NEBNext kit for Illumina (New England Biolabs, E7630S), pooled at equimolar amounts, and sequenced on a single lane of the Illumina NovaSeq 6000 S Prime flow cell (50 bp reads, paired end).

## ATAC-seq data analysis

After analyzing the quality of fastq files with fastqc (version 0.11.9), they were processed with TrimGalore (version 0.6.2), keeping the --trim-n option. Alignments to mm10 reference genome were performed with bowtie2 (*Langmead and Salzberg, 2012*) parameters: bowtie2 --very-sensitive -X 2000, and outputted in BAM format using samtools. Resultant BAM files were filtered for PCR duplicates using Picard tools, MarkDuplicates, and mitochondrial reads using samtools view (code: samtools view -h Dedup.bam | grep -v 'chrM' | samtools view `-b -h -f 0` x2 - | samtools sort > Dedup_noChrM.bam). Filtered BAM files were used to call peaks with MACS2 (parameters: -f BAMPE -n de -q 0.05 --nomodel --nolambda --keep-dup all) and generate normalized bigWig files with DeepTools, bamCoverage (options: -bs 30 --smoothLength 60 -bl --normalized using RPKM). Peak overlaps across replicates were determined using bedtools intersect (*Quinlan and Hall, 2010*). Peak calls are provided in (*Figure 4—source data 1*). Peak annotation (annotatePeaks.pl) and motif analyses (findMotifsGenome.pl) were performed with HOMER (version 4.9.1). PRDM9 motif frequency near ATAC-seq peaks was determined using HOMER annotatePeaks.pl with options: -size 20000 -hist 10. Replicate bigWig files were averaged using WiggleTools (*Zerbino et al., 2014*). Averaged bigWig files generated heatmaps with DeepTools, computeMatrix, and plotHeatmap. Data was visualized on Integrative Genomics Viewer (*Robinson et al., 2011*).

## Acknowledgements

We thank Magnuson lab members for their helpful comments on manuscript preparation. Next-generation sequencing was performed at the Duke Center for Genomic and Computational Biology. pA/G-MNase was produced by the Protein Expression and Purification core at the University of North Carolina at Chapel Hill (supported by Cancer Center Support Grant, P30 CA16086). We thank Dr. Atilla Töth (Technishe Universität Dresden) for generously providing the HORMAD1 antibody. This work was supported by National Institutes of Health grants R01GM101974 (T.M).

## Additional information

### Funding

| Funder | Grant reference number | Author |
|---|---|---|
| National Institute of General Medical Sciences | R01GM101974 | Terry Magnuson |

The funders had no role in study design, data collection and interpretation, or the decision to submit the work for publication.

### Author contributions

Debashish U Menon, Conceptualization, Data curation, Formal analysis, Validation, Investigation, Visualization, Methodology, Writing – original draft, Writing – review and editing; Prabuddha Chakraborty, Data curation, Validation, Investigation, Visualization, Methodology, Writing – review and editing; Noel Murcia, Validation, Investigation, Methodology, Writing – review and editing; Terry Magnuson, Conceptualization, Resources, Supervision, Funding acquisition, Project administration, Writing – review and editing

### Author ORCIDs

Prabuddha Chakraborty http://orcid.org/0000-0003-0775-4545
Terry Magnuson https://orcid.org/0000-0002-0792-835X

### Ethics

All animal experiments were performed according to the protocol (23.171.0-A) approved by the University of North Carolina at Chapel Hill's Institutional Animal Care and Use Committee.

Reviewer #3 (Public review): https://doi.org/10.7554/eLife.88024.5.sa1
Author response https://doi.org/10.7554/eLife.88024.5.sa2

## Additional files

### Supplementary files

- MDAR checklist

### Data availability

Sequencing data have been deposited in GEO under the accession code GSE225612. All data generated or analyzed during this study are included in the manuscript and supporting files. RNA-seq, CUT&RUN, and ATAC-seq data generated in this study are deposited with GEO (Gene Expression Omnibus) under accession number GSE225612. DMC1 SSDS (GSE35498), RAD51 SSDS (GSE143582), and H3K4me3 ChIP-seq (GSE89502) datasets were previously published (*Brick et al., 2012*; *Hinch et al., 2020*; *Maezawa et al., 2018a*; *Maezawa et al., 2018b*) and are available on GEO.

The following dataset was generated:

| Author(s) | Year | Dataset title | Dataset URL | Database and Identifier |
|---|---|---|---|---|
| Menon DU, Chakraborty P, Murcia N, Magnuson T | 2024 | ARID1A governs silencing of sex-linked transcription during male meiosis in the mouse. | http://www.ncbi.nlm.nih.gov/geo/query/acc.cgi?acc=GSE225612 | NCBI Gene Expression Omnibus, GSE225612 |

The following previously published datasets were used:

| Author(s) | Year | Dataset title | Dataset URL | Database and Identifier |
|---|---|---|---|---|
| Brick K, Smagulova F, Khil P, Camerini-Otero R, Petukhova G | 2012 | Genetic recombination is directed away from functional genetic sites in mice | http://www.ncbi.nlm.nih.gov/geo/query/acc.cgi?acc=GSE35498 | NCBI Gene Expression Omnibus, GSE35498 |
| Versari S, Longinotti G, Barenghi L, Maier J, Bradamante S | 2013 | Expression data from SPHINX (SPaceflight of Huvec: an INtegrated eXperiment) | http://www.ncbi.nlm.nih.gov/geo/query/acc.cgi?acc=GSE43582 | NCBI Gene Expression Omnibus, GSE43582 |
| Maezawa S, Hasegawa K, Kartashov A, Barski A, Namekawa S | 2018 | SCML2 establishes germline-specific bivalent domains | http://www.ncbi.nlm.nih.gov/geo/query/acc.cgi?acc=GSE89502 | NCBI Gene Expression Omnibus, GSE89502 |

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
