## [Editor Report · eLife assessment]

This study presents a **valuable** dataset regarding chromatin remodeling by the BAF complex in the context of meiotic sex chromosome inactivation. **Solid** data generally support the conclusions, although the partial deletion of the BAF complex in the germline could be considered limiting. This work will be of interest to researchers working on chromatin and reproductive biology.

---

## [Referee Report · Reviewer #3 (Public review)]

In this manuscript, Magnuson and colleagues investigate the meiotic functions of ARID1A, a putative DNA binding subunit of the SWI/SNF chromatin remodeler BAF. The authors develop a germ cell specific conditional knockout (cKO) mouse model using Stra8-cre and observe that ARID1A-deficient cells fail to progress beyond pachytene, although due to inefficiency of the Stra8-cre system the mice retain ARID1A-expressing cells that yield sperm and allow fertility. Because ARID1A was found to accumulate at the XY body late in Prophase I, the authors suspected a potential role in meiotic silencing and by RNAseq observe significant misexpression of sex-linked genes that typically are silenced at pachytene. They go on to show that ARID1A is required for exclusion of RNA PolII from the sex body and for limiting promoter accessibility at sex-linked genes, consistent with a meiotic sex chromosome inactivation (MSCI) defect in cKO mice. The authors proceed to investigate the impacts of ARID1A on H3.3 deposition genome-wide. H3.3 is known be regulated by ARID1A and is linked to silencing, and here the authors find that upon loss of ARID1A, overall H3.3 enrichment at the sex body as measured by IF failed to occur, but H3.3 was enriched specifically at transcriptional start sites of sex-linked genes that are normally regulated by ARID1A. The results suggest that ARID1A normally prevents H3.3 accumulation at target promoters on sex chromosomes and based on additional data, restricts H3.3 to intergenic sites. Finally, the authors present data implicating ARID1A and H3.3 occupancy in DSB repair, finding that ARID1A cKO leads to a reduction in focus formation by DMC1, a key repair protein. Overall the paper provides new insights into the process of MSCI from the perspective of chromatin composition and structure and raises interesting new questions about the interplay between chromatin structure, meiotic silencing and DNA repair.

In general the data are convincing. The conditional KO mouse model has some inherent limitations due to incomplete recombination and the existence of 'escaper' cells that express ARID1A and progress through meiosis normally. This reviewer feels that the authors have addressed this point thoroughly and have demonstrated clear and specific phenotypes using the best available animal model. The data demonstrate that the mutant cells fail to progress past pachytene, although it is unclear whether this specifically reflects pachytene arrest, as accumulation in other stages of Prophase is also suggested by the data in Table 1.

The revised manuscript more appropriately describes the relationship between ARID1A and DNA damage response (DDR) signaling. The authors don't see defects in a few DDR markers in ARID1A CKO cells (including a low resolution assessment of ATR), suggesting that ARID1A may not be required for meiotic DDR signaling. However, as previously noted the data do not rule out the possibility that ARID1A is downstream of DDR signaling, and the authors note the possibility of a role for DDR signaling upstream of ARID1A.

A final comment relates to the impacts of ARID1A loss on DMC1 focus formation and the interesting observation of reduced sex chromosome association by DMC1. The authors additionally assess the related recombinase RAD51 and suggest that it is unaffected by ARID1A loss. However, only a single image of RAD51 staining in the cKO is provided (Fig. S11) and there are no associated quantitative data provided. The data are suggestive and conclusions about the impacts of ARID1A loss on RAD51 must be considered as preliminary until more rigorously assessed.

Comments on latest version:

The authors have effectively addressed the minor issues raised in the most recent round of non-public reviews. This reviewer has no additional recommendations.

---

## [Author Response]

The following is the authors’ response to the previous reviews.

**Reviewer 1**:I understand that the only spermatids observed in cKO testes are coming from cells that escaped the Cre system. However, I do think that the authors could provide sperm counts data also showing decreased sperm counts in the mutant, to make their claim stronger. This is a very common fertility assessment.

All round spermatids isolated from *Arid1acKO* testes appeared only to express the normal transcript associated with the floxed allele (Fig. S4A).

[New Data - Lines 154-159] Our evaluation of the first round of spermatid development based on DNA content (1C, 2C, and 4C), revealed a significantly reduced abundance of round spermatids (1C) in mutant testes compared to wild-type testes. This finding, obtained through flow cytometry, supports the observed meiotic block at the pachytene stage (new Fig. S5A-B).

**Reviewer 3**:Lines 154-5: Currently read 'inefficient Stra8-cre inefficiency'. Should read 'inefficient Stra8-cre activity.' I see that this was noted in the first round of review but the original wording has persisted.The nucleolin antibody used should be listed in Supplementary table 3.

'inefficient Stra8-cre inefficiency' now reads “inefficient *Stra8-Cre* activity” [Line 158]

Nucleolin antibody is now listed in Supplementary Table 3